# CDE: Curiosity-Driven Exploration for Efficient Reinforcement Learning in Large Language Models

**Runpeng Dai**[1,3]**, Linfeng Song**[1]**, Haolin Liu** [1,4]**, Zhenwen Liang**[1]**, Dian Yu**[1]**, Haitao Mi**[1]**,
Zhaopeng Tu**[2]**, Rui Liu**[1,5]**, Tong Zheng**[1,5]**, Hongtu Zhu**[3]**, Dong Yu**[1]

[1]Tencent AI Lab,   [2]Tencent Multimodal Department,
[3]University of North Carolina at Chapel Hill,
[4]University of Virginia,
[5]University of Maryland, College Park
runpeng@unc.edu, lfsong@global.tencent.com

## Abstract

Reinforcement Learning with Verifiable Rewards (RLVR) is a powerful paradigm for enhancing the reasoning ability of Large Language Models (LLMs). Yet current RLVR methods often explore poorly, leading to premature convergence and entropy collapse. Moreover, they tend to produce poorly calibrated policies that remain confident in their generations regardless of correctness. To address this challenge, we introduce **Curiosity-Driven Exploration (CDE)**, a framework that leverages the model's intrinsic sense of curiosity to guide exploration. We formalize curiosity with signals from both the actor and the critic: for the actor, we use perplexity [1] over its generated response, and for the critic, we use the variance of value estimates from a multi-head critic architecture. Both signals serve as an exploration bonus within the RLVR framework to guide the model. Our theoretical analysis shows that the actor-wise bonus inherently penalizes overconfident errors and promotes diversity among correct responses; moreover, we connect the critic-wise bonus to the well-established count-based exploration bonus in RL. Empirically, our method achieves an approximate **+3** point improvement over standard RLVR using GRPO/PPO on AIME benchmarks.

## 1 Introduction

Reinforcement learning has significantly advanced the reasoning capabilities of Large Language Models (LLMs). A leading paradigm is Reinforcement Learning with Verifiable Rewards (RLVR), which directly optimizes models based on the correctness of their final answers. Despite the development of advanced RLVR algorithms like GRPO (Guo et al., 2024) and DAPO (Yu et al., 2025b), critical challenges persist. Problems such as premature convergence and entropy collapse (Cui et al., 2025) are frequently observed, posing fundamental challenges to the efficiency of RLVR.

These challenges stem from the classic exploration-exploitation dilemma in reinforcement learning (Sutton & Barto, 2018). Phenomena like entropy collapse reveal a critical flaw in the training process: it is heavily biased towards exploitation, causing models to converge prematurely instead of sufficiently exploring their environment for better solutions. Although the RL literature encompasses a wide range of exploration strategies, these methods exhibit limitations when applied to LLMs. Simple heuristics, including $\epsilon$-greedy policies (Sutton & Barto, 2018) and entropy bonuses (Haarnoja et al., 2018), either inject randomness to the environment or encourage the policy to be more stochastic. Directly applying those approaches often demonstrates debatable effectiveness in complex environments like Deep RL (Andrychowicz et al., 2021) and LLM-based reasoning (Cui et al., 2025; Shen, 2025).

---

[1]Our bonus is technically the logarithm of the perplexity (log-PPL). For simplicity, we refer to it as the **PPL bonus** throughout the paper.

More principled strategies include count-based and prediction-based approaches. The former, including UCB (Lai, 1987) and LinUCB (Li et al., 2010), incentivizes visits to rarely explored states, while the latter, like ICM (Pathak et al., 2017) and RND (Burda et al., 2018b), reward an agent for reaching hard-to-predict states. Recent work has focused on adapting these methods for tuning LLMs (Bai et al., 2025; Sun et al., 2025; Gao et al., 2025; Yu et al., 2025a). However, these strategies often require training auxiliary modules and effective state representations (Burda et al., 2018a). This requirement is particularly challenging for LLMs, where efficiently represent a reasoning path into a fixed-size embedding remains an open problem (Fu et al., 2024), and simplistic approaches such as using the last hidden state are often problematic (Barbero et al., 2024).

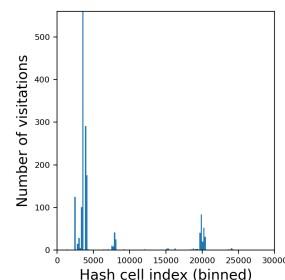

We first explored count-based methods using the SimHash technique (Tang et al., 2017), which maps the embedding of a CoT response into a discrete hash code, and then uses the visitation frequency of hash cells as pseudo-counts (see Appendix D). However, this approach proved problematic, exposing the inherent challenge of representing complex reasoning trajectories. As illustrated in Figure 1, most responses collapsed into a small number of hash grids. This clustering creates a highly concentrated count distribution, undermining the effectiveness of count-based exploration. Therefore, developing efficient and scalable exploration methods for LLMs remains a critical challenge.

Figure 1: Distribution of number of visitations across hash-cells.

In this work, we propose an intuitive approach that leverages the model's intrinsic sense of **curiosity** as a guide for exploration. An LLM, having been trained on vast reasoning corpora, develops a sophisticated internal model of what constitutes a familiar versus a novel reasoning pattern. This parallels early childhood development (Chu & Schulz, 2020), where learning is not driven by an external summary and count of experiences, but is instead propelled by an intrinsic curiosity to explore novel situations. We formalize this principle in our **Curiosity-Driven Exploration (CDE)** framework, which considers curiosity signals from both the actor and the critic. For the actor, perplexity (PPL) over its generated response serves as the curiosity measure. For the critic, we measure curiosity via the variance of its posterior value distribution. We then approximate this posterior by extending the PPO framework with a multi-head, bootstrapped structure (Figure 2).The curiosity signals serve as an exploration bonus, shaping the reward and advantage functions to effectively guide exploration.

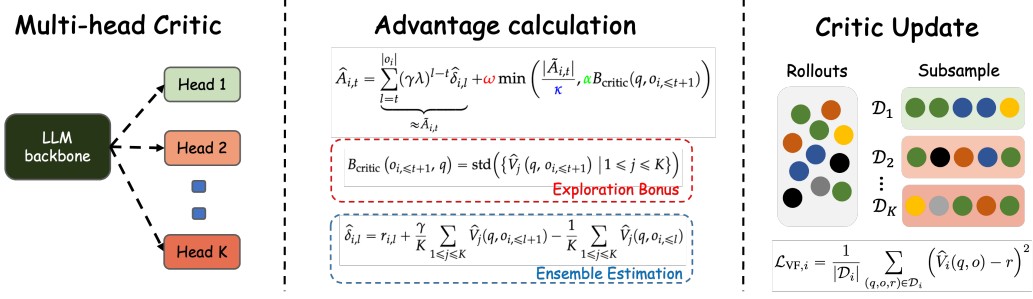

Figure 2: Illustration of the multi-head critic framework.

We provide a theoretical foundation for our method, establishing two key properties. First, we prove that the actor's perplexity bonus intrinsically penalizes overconfident errors while encouraging diversity among correct responses (Theorem 3.1). Second, we connect the critic's bonus to classical RL theory by showing it is equivalent to count-based bonuses in the linear MDP setting (Theorem 3.2), grounding our approach in established exploration principles.

Empirically, our method delivers consistent performance gains across four widely used mathematics benchmarks (AIME25, AIME24, AMC23, and MATH), achieving a notable **+3** point improvement on the challenging AIME benchmarks. Beyond these performance gains, our analysis of the training dynamics uncovers a critical failure mode we term **calibration collapse**: under a standard GRPO policy, a model's confidence progressively decouples from its correctness, while the proposed PPL bonus mitigates this miscalibration.

## 2 PRELIMINARIES: RLVR, GRPO AND PPO

We frame the language generation process as a sequential decision-making problem, trained using the RLVR paradigm (Lambert et al., 2024; Yu et al., 2025b). In RLVR, a rule-based verifier assesses the correctness of a generated response by comparing it against the ground truth, providing a verifiable reward signal.

**Group Relative Policy Optimization** (GRPO (Shao et al., 2024)) is a critic-free RL algorithm. Given a prompt $q$, the policy $\pi_\theta$ generates a group of G candidate responses $\{o_1, o_2, \ldots, o_G\}$. Each response $o_i$ receives a reward $r_i = r(o_i, q)$. The advantage of the $i$-th response is calculated by normalizing its reward relative to the others in the group: $A_i = (r_i - \text{mean}(r_1, \ldots, r_G))/\text{std}(r_1, \ldots, r_G)$. Let $\pi_{\theta_{\text{old}}}$ be the behavior policy used to generate the responses. The policy is updated by maximizing the GRPO objective:

$$\mathcal{J}_{\text{GRPO}}(\theta) = \mathbb{E}_{q \sim \mathcal{D}, \{o_i\}_{i=1}^G \sim \pi_{\theta_{\text{old}}}} \left[ \frac{1}{G} \sum_{i=1}^{G} \frac{1}{|o_i|} \sum_{t=1}^{|o_i|} \min \left( \tilde{r}_{i,t} A_i, \text{clip}(\tilde{r}_{i,t}, 1 - \varepsilon, 1 + \varepsilon) A_i \right) \right],$$

where $\tilde{r}_{i,t} = \pi_\theta(o_{i,t} \mid q, o_{i,<t})/\pi_{\theta_{\text{old}}}(o_{i,t} \mid q, o_{i,<t})$ is the token-level probability ratio.

**Proximal Policy Optimization** (PPO(Schulman et al., 2017)) is an actor-critic algorithm that uses a learned value function, or critic $V_\phi$, to estimate advantages. For a single response $o$ with a final reward $r$, the advantage at token $t$ is:

$$A_t = \sum_{l=t}^{|o|} (\gamma\lambda)^{l-t} \delta_l, \quad \text{where} \quad \delta_l = r_l + \gamma V_\phi(q, o_{\leqslant l+1}) - V_\phi(q, o_{\leqslant l}).$$

Here, the reward is applied only at the final token ($r_{|o|} = r$ and $r_l = 0$ for $l < |o|$). The hyperparameters $\gamma$ and $\lambda$ are the discount factor and GAE trace-decay parameter, respectively. The PPO policy objective is:

$$\mathcal{J}_{\text{PPO}}(\theta) = \mathbb{E}_{q \sim \mathcal{D}, o \sim \pi_{\theta_{\text{old}}}} \left[ \frac{1}{|o|} \sum_{t=1}^{|o|} \min \left( \tilde{r}_t A_t, \text{clip}(\tilde{r}_t, 1 - \varepsilon, 1 + \varepsilon) A_t \right) \right].$$

## 3 CDE: CURIOSITY-DRIVEN EXPLORATION

This section first analyzes count-based exploration, highlighting a key challenge in its application. We then introduce Curiosity-Driven Exploration (CDE), a systematic framework that leverages curiosity signals from both the actor and the critic. We elaborate on the specific formulations for actor and critic curiosity in Sections 3.2 and 3.3, respectively.

### 3.1 CHALLENGE OF COUNT-BASED EXPLORATION FOR RLVR

Count-based exploration methods use response embeddings to quantify novelty, assigning an exploration bonus to rare or less frequent responses patterns. To implement this efficiently, we investigated a hash-based counting method (Tang et al., 2017), which projects CoT embeddings into discrete hash grids and uses grid visitation frequency as pseudo-counts (details in Appendix D).

While conceptually appealing, this approach fails in practice. We hypothesize the root cause is the **poor expressiveness of embeddings**. Effectively representing a reasoning trajectory in a single embedding is challenging, and simple heuristics such as the last token's hidden state, are often problematic (Barbero et al., 2024). As illustrated in Figure 1, where most responses collapse into the same or neighboring hash grids, leading to a highly concentrated distribution of counts and thus undermining the effectiveness of count-based exploration for RLVR.

In this work, we introduce an exploration paradigm guided by the model's intrinsic **curiosity**, moving beyond methods that rely on explicit counts or response embeddings. Our approach is inspired by early cognitive development (Chu & Schulz, 2020), where, much like children, the model is encouraged to explore what it finds novel while acting confidently on familiar patterns. *Learning is therefore not driven by an external count of experiences but is propelled by an intrinsic drive to explore.* We formalize this principle with separate curiosity signals for the actor and the critic, beginning with the actor's implementation.

## 3.2 EXPLORATION GUIDED BY ACTOR CURIOSITY

We model actor curiosity as the actor's uncertainty about its own actions. Intuitively, a response that is surprising to the actor—i.e., has a low probability under its current policy—likely resides in an underexplored region of its learned distribution.

A natural and computationally efficient measure of this surprise is the perplexity of the actor's generation. We formalize this as a response-level curiosity bonus, defined as the negative average log-probability of a generated response $o = \{o_1, \ldots, o_T\}$, given a prompt $q$:

$$B_{\text{actor}}(q, o) = -\frac{1}{T} \sum_{t=1}^{\top} \log \pi(o_t | o_{<t}, q) \tag{1}$$

where $\pi$ denotes the actor policy. A higher value for $B_{\text{actor}}(q, o)$ indicates greater surprise and thus a stronger intrinsic reward signal for exploration.

However, simply adding this bonus to the original reward can be unstable and sub-optimal. Unconstrained exploration might incentivize the model to generate high-perplexity but low-quality or inaccurate responses (a behavior known as reward hacking), or lead to over-exploration where the policy fails to converge to a stable, high-quality output. To address this, we integrate the bonus using an adaptive clipping mechanism. The total response-level reward, $\widehat{r}$, is a combination of the original reward signal $r(q, o)$:

$$\widehat{r}(q, o) = r(q, o) + \omega \min\left(\frac{|r(q, o)|}{\kappa}, \alpha B_{\text{actor}}(q, o)\right) \tag{2}$$

This formulation promotes exploration by rewarding responses that the actor finds surprising, while constraining the bonus to remain a fraction of the original reward. The behavior of this reward function is controlled by three key hyperparameters:

- The **bonus weight** $\omega$ is a dynamic coefficient, typically set with an annealing schedule to decrease over the course of training. This allows for aggressive exploration in the early stages and then gradually shifts focus towards exploitation of high-reward regions as the policy converges.
- The **clipping ratio** $\kappa$ governs the maximum size of the curiosity bonus relative to the original reward. By capping the bonus at $|r(q, o)|/\kappa$, it ensures the bonus remains a supplement and prevents it from dominating the learning signal.
- The **bonus scaling factor** $\alpha$ normalizes the curiosity bonus $B_{\text{actor}}(q, o)$ before it is compared to the clipped reward. A higher $\alpha$ allows the curiosity bonus to reach the clipping threshold more easily, whereas a smaller $\alpha$ diminishes its potential impact.

**Calibration Effect**  Beyond exploration, we study the *calibration* effect of the PPL bonus. To build this intuition, we analyze responses along two axes: correctness and PPL. Among these four categories, two require particular attention:

1. Incorrect responses with low PPL indicate that the model is highly confident in its answer, yet the response is wrong. This reflects overfitting and should be penalized.
2. Correct responses with high PPL suggest that the model is less familiar with such answers, but they nevertheless turn out to be successful. This reflects effective exploration and should be encouraged.

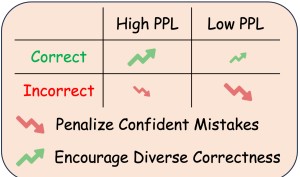

Figure 3: Responses by correctness and PPL.

As illustrated in Figure 3, the PPL bonus intrinsically penalizes confident mistakes while encouraging novel correct responses. For correct responses, those novel responses (with higher PPL) receive a larger positive reward. For incorrect responses, those confident responses (with lower PPL) receive larger penalty as it receives smaller PPL bonus. The following theorem formalizes this intuition; its precise statement and proof are deferred to Appendix F.

**Theorem 3.1.** *Let $\pi_h$ denote the policy at training step $h$. With PPL bonus in equation 1, the update to $\pi_{h+1}$ calibrates the policy's confidence as follows:*

*(i) Among correct responses, those with higher PPL receive a larger relative probability increase.*

*(ii) Among incorrect responses, those with lower PPL receive a larger relative probability decrease.*

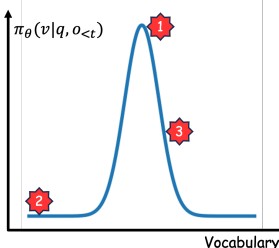

Figure 4: The next-token distribution of the policy $\pi_\theta$.

The PPL bonus can be viewed as an enhancement to the traditional entropy based signals by incorporating sample-specific information. The entropy bonus is inherently sample-agnostic; as defined in Equation 3, the entropy $\mathcal{H}_t$ is computed over the entire next-token probability distribution, $\pi_\theta\left(v \mid q, o_{<t}\right)$, making it independent of the token that is actually sampled. As illustrated in Figure 4, this means that even when the model makes a high-confidence error (e.g., by sampling Token 1), the entropy bonus fails to penalize that choice.

$$\mathcal{H}_t = -\sum_{v \in \mathcal{V}} \pi_\theta\left(v \mid q, o_{<t}\right) \log \pi_\theta\left(v \mid q, o_{<t}\right). \tag{3}$$

### 3.3 EXPLORATION GUIDED BY CRITIC CURIOSITY

In contrast to critic-free methods such as REINFORCE and GRPO, the critic (value function) in actor–critic frameworks provides a higher-level understanding of the prompt–response pair by estimating the expected reward-to-go. Since this estimate is learned directly from collected trajectories, its posterior distribution conditioned on the observed data naturally reflects the degree of coverage: regions with dense data yield concentrated (low-variance) posteriors, whereas sparsely sampled regions result in higher uncertainty. Posterior distributions are a well-established means of quantifying predictive uncertainty in deep learning models (Gal & Ghahramani, 2016; Lakshminarayanan et al., 2017; Yu et al., 2025a). As shown in Figure 5, the orange curve exhibits lower variance—evidence of better data coverage—whereas the other curve is more dispersed, reflecting greater uncertainty.

To approximate the posterior distribution of value estimates, we adopt the classical bootstrap method (Davison & Hinkley, 1997), widely used in statistics and increasingly recognized in the RL community as an effective tool for exploration (Osband et al., 2016; Ciosek et al., 2019; Bai et al., 2021). We implement this idea through a multi-head critic (left subfigure in Figure 2), where $K$ critics $\{\widehat{V}_1, \ldots, \widehat{V}_K\}$ share a common LLM backbone. Each head is trained on a resampled subset of the collected trajectories (right subfigure in Figure 2), thereby producing an empirical approximation to the posterior distribution.

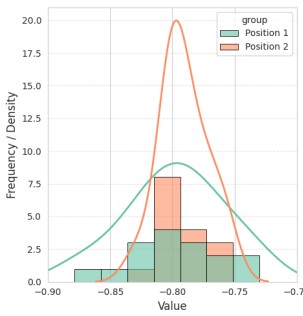

Figure 5: An illustration of two posterior distributions of the critic and their bootstrap approximations.

We use the standard deviation across the $K$ heads as a principled curiosity signal to guide the policy toward under-explored regions where the heads have high disagreement. In the following theorem, we establish a surprising yet intuitive result: under a Linear MDP assumption, this standard deviation is a consistent estimator of the pseudo-count bonus.

**Theorem 3.2.** *In linear MDPs, the standard deviation across multi-head critics is a consistent estimator of the pseudo-count exploration bonus, $\sqrt{\phi_{n,t}^\top \Lambda_{n,t}^{-1} \phi_{n,t}}$, as used in LSVI-UCB (Jin et al., 2020) and CFPO (Cassel & Rosenberg, 2024), where $\phi_{n,t} = \phi(s_{n,t}, a_{n,t})$ denotes the feature of state-action pair and $\Lambda_{n,t} = \sum_{i=0}^{n} \phi_{i,t}\phi_{i,t}^\top + \lambda I$ is the coverage matrix.*

Appendix G provides the rigorous formulation of the linear MDP assumptions and the proof for Theorem 3.2, with further empirical support in Section 4.4. Building on this foundation, our multi-head PPO algorithm adapts the standard PPO training stages: (i) generating trajectories with the actor, (ii) updating the actor, and (iii) updating the critic. The key distinction is our incorporation of multi-head variance as an exploration bonus. The full procedure is detailed below and illustrated in Figure 2.

- **Actor roll-out:** For clarity, we describe the process for a single prompt $q$. The actor generates a set of n responses, $\{o_1, \ldots, o_n\}$, each of which receives a corresponding verifiable reward $r_i$.
- **Actor update:** In this step, the advantage is calculated as:

$$\widehat{A}_{i,t} = \underbrace{\sum_{l=t}^{|o_i|} (\gamma\lambda)^{l-t} \widehat{\delta}_{i,l}}_{\approx \tilde{A}_{i,t}} + \textcolor{red}{\omega} \min\left(\frac{|\tilde{A}_{i,t}|}{\textcolor{blue}{\kappa}}, \alpha B_{\text{critic}}(q, o_{i,\leqslant t+1})\right).$$

The advantage consists of two components. The first term, $\tilde{A}_{i,t}$, is a modified PPO advantage estimate that leverages an *ensemble* of K value heads instead of a single value function:

$$\widehat{\delta}_{i,l} = r_{i,l} + \frac{\gamma}{K} \sum_{j=1}^{K} \widehat{V}_j(q, o_{i,\leqslant l+1}) - \frac{1}{K} \sum_{j=1}^{K} \widehat{V}_j(q, o_{i,\leqslant l}).$$

The second term introduces the *multi-head critic bonus* ($B_{\text{critic}}$), governed by the bonus weight $\omega$, clipping ratio $\kappa$, and scaling factor $\alpha$ (see discussion following Equation 2 for interpretation). Specifically, $B_{\text{critic}}$ is defined as the standard deviation across the $K$ value heads, encouraging exploration by assigning higher bonus to actions leading to uncertain/less-visited regions:

$$B_{\text{critic}}(q, o_{i,\leqslant t+1}) = \text{std}\left(\left\{\widehat{V}_j(q, o_{i,\leqslant t+1}) \big| 1 \leqslant j \leqslant K\right\}\right).$$

- **Critic update:** The critic heads are updated using the collected roll-outs, which form a dataset $\mathcal{D} = \{(q, o_{i,\leqslant t}, r_i) | i \in [n], t \in [|o_i|]\}$. Each critic head $j$ is trained on a subset of the data, $\mathcal{D}_j \subset \mathcal{D}$, sampled with replacement. The hyperparameter $\zeta \in (0, 1]$ controls the size of this subset $|\mathcal{D}_j| = \zeta|\mathcal{D}|$, creating a trade-off: a smaller $\zeta$ increases head diversity, while a larger $\zeta$ improves sample efficiency. The multi-head critic is then trained by minimizing the following bootstrap loss:

$$\mathcal{L}_\phi = \frac{1}{\zeta K |\mathcal{D}|} \sum_{j=1}^{K} \sum_{(q,o,r) \in \mathcal{D}_j} \left(\widehat{V}_j(q, o) - r\right)^2.$$

## 4 EXPERIMENTS

### 4.1 DATASET AND MODEL

We evaluate our proposed method, CDE, against standard PPO and GRPO baselines on four challenging mathematical reasoning benchmarks: MATH (Hendrycks et al., 2021), AMC23 (MAA, b), AIME24, and AIME25 (MAA, a). All experiments are implemented in the Verl framework, using the Qwen3-4B-Base model (Yang et al., 2025) and Llama-3.2-3B-Instruct (Grattafiori et al., 2024) trained on the DAPO-17K dataset (Yu et al., 2025b). Full implementation details and the prompt are provided in Appendix B.

### 4.2 MAIN RESULTS

We benchmark our method against the original GRPO/PPO as well as two additional baselines: the entropy bonus for traditional entropy-based exploration (Schulman et al., 2017) and i-MENTOR for curiosity-driven exploration (Gao et al., 2025). The main results are presented in Table 1 while the training dynamics is presented in Figure 6. Here $K$ Heads represents multi-head critic PPO with $K$ heads. For all evaluations, we set the sampling temperature to 1.0. On the MATH dataset, we report accuracy from a single response (Avg@1). Following the protocol of (Wang et al., 2025), for the other three datasets we generate 16 independent samples and report both Pass@16 and the average accuracy (Avg@16). The key observations are as follows:

- The PPL bonus enhances the mathematical reasoning ability of the GRPO method, yielding an improvement of approximately +2 points across datasets and models. In particular, our method achieves notable gains on Pass@16, surpassing the baseline GRPO over Qwen3-4B-Base model by about +8 points on the AIME24 dataset.

Table 1: Zero-shot accuracy on the validation datasets. **Avg** column represents the overall accuracy across datasets, calculated by averaging the Avg@1 score from MATH with the Avg@16 scores (mean and standard deviation) from the remaining datasets. We use "–" to exclude the results for Llama-3.2-3B-Instruct on AIME25, since its performance is approximately 1 point.

| Model | MATH Avg@1 | AMC23 Avg@16 | AMC23 Pass@16 | AIME24 Avg@16 | AIME24 Pass@16 | AIME25 Avg@16 | AIME25 Pass@16 | Avg |
|---|---|---|---|---|---|---|---|---|
| *GRPO based methods* | | | | | | | | |
| Qwen3-4B-Base-GRPO | 87.3 | $63.6 \pm 0.9$ | 89.1 | $20.8 \pm 1.1$ | 41.9 | $21.0 \pm 0.8$ | 39.2 | 48.2 |
| ↳ w/ Entropy bonus | 86.8 | $64.3 \pm 0.6$ | 89.7 | $21.8 \pm 1.0$ | 39.4 | $21.2 \pm 0.4$ | 41.1 | 48.5 |
| ↳ i-MENTOR-GRPO | 87.6 | $63.2 \pm 0.7$ | 89.1 | $22.5 \pm 0.3$ | 39.3 | $23.0 \pm 0.4$ | 40.4 | 49.1 |
| ↳ **w/ PPL bonus** | 87.7 | $67.8 \pm 1.3$ | 89.5 | $23.3 \pm 0.9$ | 48.5 | $23.5 \pm 1.0$ | 42.5 | 50.6 |
| | | | | | | | | |
| Llama-3.2-3B-Instruct-GRPO | 53.8 | $30.0 \pm 0.3$ | 59.4 | $13.1 \pm 0.2$ | 30.7 | - | - | 32.3 |
| ↳ **w/ PPL bonus** | 55.8 | $32.8 \pm 0.5$ | 65.0 | $14.6 \pm 0.3$ | 35.1 | - | - | 34.4 |
| | | | | | | | | |
| *PPO based methods* | | | | | | | | |
| Qwen3-4B-Base-PPO | 86.6 | $64.1 \pm 0.6$ | 87.2 | $17.8 \pm 0.5$ | 36.0 | $17.5 \pm 0.8$ | 33.7 | 46.5 |
| ↳ w/ Entropy bonus | 83.9 | $66.3 \pm 1.2$ | 87.5 | $17.9 \pm 0.8$ | 30.6 | $19.5 \pm 0.4$ | 33.0 | 46.9 |
| ↳ i-MENTOR-PPO | 85.8 | $60.9 \pm 0.9$ | 85.9 | $18.9 \pm 0.5$ | 39.4 | $19.6 \pm 0.7$ | 36.5 | 46.3 |
| ↳ **w/ PPL bonus** | 87.9 | $66.1 \pm 1.0$ | 88.5 | $18.3 \pm 0.5$ | 37.6 | $18.3 \pm 1.1$ | 33.5 | 47.7 |
| ↳ **w/ 2 Heads** | 83.2 | $63.6 \pm 0.9$ | 89.9 | $19.6 \pm 0.7$ | 34.8 | $19.6 \pm 1.0$ | 36.1 | 46.6 |
| ↳ **w/ 4 Heads** | 87.3 | $63.9 \pm 1.2$ | 87.9 | $21.5 \pm 0.7$ | 35.5 | $21.5 \pm 1.3$ | 45.5 | 48.5 |
| ↳ **w/ 8 Heads** | 85.1 | $66.7 \pm 0.8$ | 86.9 | $21.7 \pm 1.3$ | 46.4 | $19.0 \pm 0.8$ | 37.1 | 48.1 |
| ↳ **w/ 16 Heads** | 88.3 | $65.0 \pm 0.9$ | 88.7 | $20.5 \pm 0.8$ | 41.9 | $20.0 \pm 1.3$ | 38.8 | 48.6 |
| | | | | | | | | |
| Llama-3.2-3B-Instruct-PPO | 51.9 | $27.2 \pm 0.2$ | 59.1 | $12.5 \pm 0.3$ | 29.2 | - | - | 30.5 |
| ↳ **w/ 4 Heads** | 56.1 | $32.2 \pm 0.4$ | 63.7 | $14.3 \pm 0.1$ | 35.5 | - | - | 34.2 |

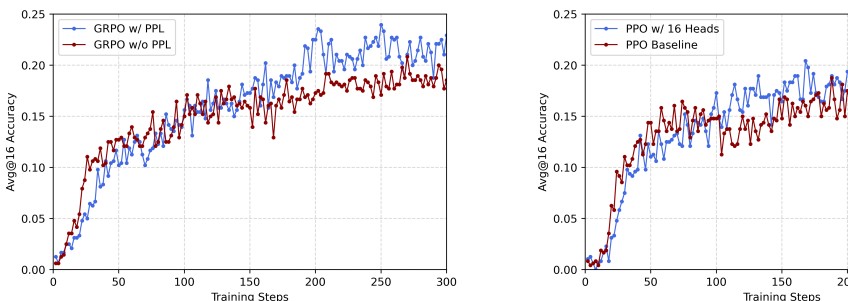

Figure 6: Comparison of Avg@16 accuracy on AIME25 over training of vanilla GRPO and PPO (Baseline methods) and GRPO with PPL bonus and 16 head multi-head PPO (Our methods).

- Across benchmarks, multi-head PPO consistently outperforms vanilla PPO. Using $K = 4$ and $K = 16$ heads yields average gains of roughly $+2$ points, and we observe an increase of around $+10$ points in Pass@16 on AIME datasets with Qwen3-4B-Base model in many cases.

- The performance of multi-head PPO generally increases with the number of heads $K$: with $K = 2$ delivers negligible gains over the baseline, and performance begins to plateau once $K \geqslant 4$, which suggests that a modest number of heads can capture most of the curiosity signals needed.

- The consistent outperformance of multi-head PPO ($K \geqslant 4$) over PPO with a PPL bonus suggests that the critic's disagreement reflects a higher-level uncertainty about the long-term value of a reasoning path, which provides a more strategic exploration signal.

- Figure 6 illustrates CDE's effective exploration strategy: while initially lagging the PPO/GRPO baselines in test accuracy, it achieves superior final performance. This initial trade-off prevents the model from prematurely exploiting spurious high-reward paths. As exploration expands state-action coverage, the policy refines, enabling a more effective shift to exploitation that yields a higher accuracy ceiling.

We further provide an analysis of the multi-head critic's computational cost, including memory usage and runtime, in Appendix C.

### 4.3 UNDERSTANDING THE EFFECT OF THE PPL BONUS

***Bonus weight decay is crucial.*** We compare four schedules for the bonus weight $\omega$—*No decay*, *Linear*, *Cosine*, and *Staircase*—as illustrated in Figure 7, with the performance of models trained under each schedule summarized in Table 2. Briefly, the *No decay* schedule maintains strong exploration throughout training, while the *Staircase* schedule reduces $\omega$ abruptly, enabling strong exploration in the early phase and then removing the bonus for final convergence. The *Linear* and *Cosine* schedules provide intermediate behaviors.

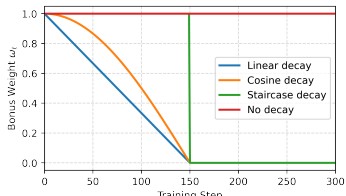

Figure 7: An illustration of different weight anneal schedules.

The results in Table 2 underscore two insights: First, decay of the bonus weight is necessary, as all decay schedules outperform the no-decay baseline. Second, strong exploration in the early phase is crucial, with the staircase scheme proving most effective by sustaining high exploration initially to broaden state–action coverage and then removing the bonus abruptly to allow stable convergence. We also track entropy dynamics, providing extra support for the importance of weight decay (see Figure 11, Appendix E).

Table 2: Zero-shot accuracy of GRPO models under different PPL bonus weight decay schedules. The schedules follow those illustrated in Figure 7.

| Model | MATH | AMC23 | | AIME24 | | AIME25 | | Avg |
|---|---|---|---|---|---|---|---|---|
| | Avg@1 | Avg@16 | Pass@16 | Avg@16 | Pass@16 | Avg@16 | Pass@16 | |
| *Bonus Weight Decay Schedules* | | | | | | | | |
| Qwen3-4B-Base-GRPO | 87.3 | 63.6 | 91.1 | 21.0 | 41.9 | 20.8 | 39.2 | 48.2 |
| ↳ $\omega$ **No decay** | 85.1 | 64.5 | 84.6 | 20.8 | 39.0 | 22.3 | 36.2 | 48.2 |
| ↳ $\omega$ **Linear decay** | 85.4 | 66.1 | 91.9 | 23.3 | 40.4 | 20.0 | 40.4 | 48.7 |
| ↳ $\omega$ **Cosine decay** | 86.7 | 68.1 | 90.0 | 22.5 | 44.9 | 21.5 | 40.7 | 49.7 |
| ↳ $\omega$ **Staircase decay** | 87.7 | 67.8 | 89.2 | 23.5 | 48.5 | 23.3 | 40.3 | 50.6 |

***Analysis of Calibration.*** As shown in Figure 8(a), models trained with standard GRPO suffer from a phenomenon we term **calibration collapse**: early in standard GRPO training, correct responses have lower PPL (higher confidence) than incorrect ones, but as training progresses this gap shrinks. By contrast, with a PPL bonus (subfigure (b)), this separation is sustained throughout training. This pattern can be partially explained by Theorem 3.1: while both standard GRPO and GRPO with a PPL bonus tend to increase confidence on correct answers, the PPL bonus suppresses confident errors, thereby improving calibration.

This finding is original and practically important. Ideally, a trained model should be faithful—confident when its answer is correct and cautious when it is not. Better calibration enhances interpretability and supports inference-time selection strategies such as self-certainty BoN (Wang et al., 2022) and DeepConf (Fu et al., 2025). It also connects to the growing literature on calibrating LLMs, both during training (e.g., (Shen et al., 2024)) and at test time (e.g., (Ulmer et al., 2024)).

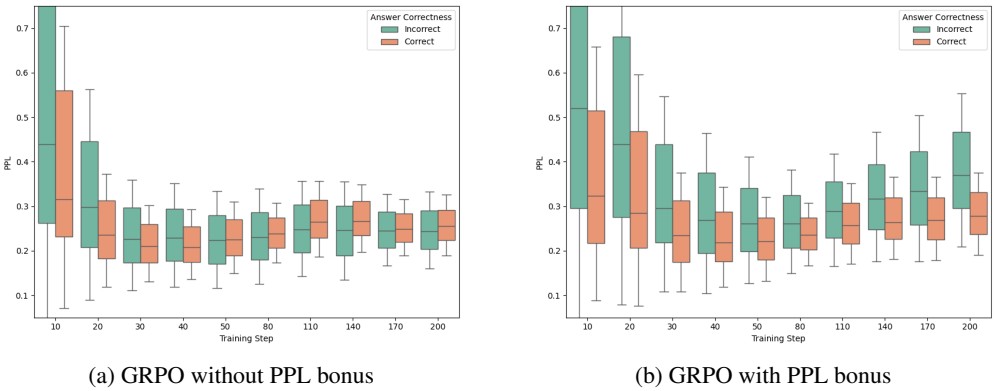

(a) GRPO without PPL bonus

(b) GRPO with PPL bonus

Figure 8: Average response PPL per training step, stratified by correctness.

### 4.4 FURTHER ANALYSIS OF THE MULTI-HEAD CRITIC

***Analysis of sub-sample fraction $\zeta$.*** Additionally, we examine the sensitivity of the critic update to the hyperparameter $\zeta$ (sub-sample fraction). We vary $\zeta$ under two configurations—critics with 16 heads and with 4 heads—and compare $\zeta \in \{0.5, 1\}$. As shown in Table 3, while a smaller number of heads benefits from a larger sub-sample fraction, the overall performance is stable across settings. The model demonstrates robustness to the masking fraction $\zeta$, achieving similar results for both values tested (0.5 and 1.0).

Table 3: Ablation study on sub-sample fraction $\zeta$.

| Model | MATH | AMC23 | | AIME24 | | AIME25 | | Avg |
|---|---|---|---|---|---|---|---|---|
| | Avg@1 | Avg@16 | Pass@16 | Avg@16 | Pass@16 | Avg@16 | Pass@16 | |
| *Mask fraction* | | | | | | | | |
| **16 Heads ; $\zeta = 0.5$** | 88.3 | 65.0 | 88.7 | 20.5 | 41.9 | 20.0 | 38.8 | 48.6 |
| 16 Heads ; $\zeta = 1$ | 85.4 | 65.3 | 85.3 | 21.0 | 39.2 | 21.7 | 43.2 | 48.4 |
| 4 Heads ; $\zeta = 0.5$ | 86.1 | 66.4 | 85.8 | 18.1 | 36.7 | 23.1 | 39.1 | 48.4 |
| **4 Heads ; $\zeta = 1$** | 87.3 | 63.9 | 87.9 | 21.5 | 35.5 | 21.5 | 45.5 | 48.5 |

**Analysis of dynamics of $B_{\text{critic}}$** across different datasets and over the course of training. First, we find that $B_{\text{critic}}$ is sensitive to data novelty. As shown in Figure 9, we compared the average critic disagreement across three datasets. The disagreement is lowest on the familiar training data (DAPO-17K) and significantly higher on unseen data from both an in-domain validation set (AMC23) and an out-of-domain set (GPQA). This result supports the intuition that the multi-head critics tend to show stronger disagreement on data that is less frequently encountered during training.

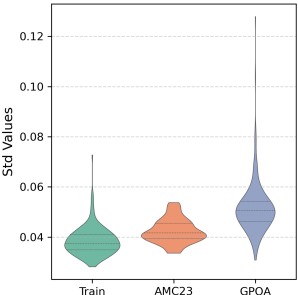

Figure 9: Distribution of the std of value heads across different datasets.

Second, the exploration bonus naturally anneals over the course of training. As the model repeatedly encounters similar reasoning paths, the critics' value estimates converge, thus reducing the bonus. This dynamic provides empirical support for interpreting the multi-head bonus as an implicit, count-based exploration mechanism (see Figure 8, Appendix E).

### 4.5 SENSITIVITY ANALYSIS

In this subsection, we conduct a more comprehensive ablation study on the hyperparameters $\kappa$ and $\alpha$ in Equation 2.

Table 4: Sensitivity analysis of Qwen3-4B-Base-GRPO under different $(\kappa, \alpha)$ settings.

| Model | MATH | AMC23 | | AIME24 | | AIME25 | | Avg |
|---|---|---|---|---|---|---|---|---|
| | Avg@1 | Avg@16 | Pass@16 | Avg@16 | Pass@16 | Avg@16 | Pass@16 | |
| Qwen3-4B-Base-GRPO | 87.3 | 63.6 | 89.1 | 20.8 | 41.9 | 21.0 | 39.2 | 48.2 |
| $\rightarrow \kappa = 3, \alpha = 1$ (Original) | 87.7 | 67.8 | 89.5 | 23.3 | 48.5 | 23.5 | 42.5 | 50.6 |
| $\rightarrow \kappa = 1, \alpha = 1$ | 20.9 | 11.1 | 54.6 | 1.5 | 12.7 | 1.3 | 11.4 | 8.67 |
| $\rightarrow \kappa = 2, \alpha = 1$ | 87.6 | 66.5 | 86.9 | 22.3 | 42.5 | 21.6 | 40.1 | 49.5 |
| $\rightarrow \kappa = 4, \alpha = 1$ | 88.6 | 68.2 | 90.1 | 23.6 | 46.6 | 22.7 | 40.8 | 50.8 |

Table 5: Sensitivity analysis of Qwen3-4B-Base-PPO under different $(\kappa, \alpha)$ settings.

| Model | MATH | AMC23 | | AIME24 | | AIME25 | | Avg |
|---|---|---|---|---|---|---|---|---|
| | Avg@1 | Avg@16 | Pass@16 | Avg@16 | Pass@16 | Avg@16 | Pass@16 | |
| Qwen3-4B-Base-PPO | 86.6 | 64.1 | 87.2 | 17.8 | 36.0 | 17.5 | 33.7 | 46.5 |
| $\rightarrow \kappa = 3, \alpha = 0.5$ (Original) | 87.3 | 63.9 | 87.9 | 21.5 | 35.5 | 21.5 | 45.5 | 48.5 |
| $\rightarrow \kappa = 1, \alpha = 1$ | 86.5 | 65.0 | 89.5 | 20.8 | 42.5 | 20.5 | 39.8 | 48.2 |

As illustrated in Table 5, the critic-wise bonus is generally robust to hyperparameter choices; setting both $\kappa$ and $\alpha$ to 1 produces nearly identical performance. The actor-wise bonus is relatively more

sensitive. Generally, to get relative good performance, the clipping fraction $\kappa$ should not be set to small, i.e. the bonus should be constrained to be not exceeding at least 50% of the labeled reward.

## 5 RELATED WORK

### 5.1 REINFORCEMENT LEARNING (RL) FOR LLM REASONING

Reinforcement Learning is a central technique for advancing the reasoning capabilities of LLMs. Initial approaches relied on reward models that provided either outcome-based supervision, focusing on the final answer (Cobbe et al., 2021), or process-based supervision (Uesato et al., 2022). More recently, RLVR (Lambert et al., 2024) has emerged as a powerful alternative, demonstrating significant performance on complex reasoning tasks in mathematics and coding (Guo et al., 2025). Parallel efforts aim to improve upon the standard RLVR paradigm with techniques such as self-evolving (Huang et al., 2025; He et al., 2025), parallel thinking (Zheng et al., 2025; 2026). Despite these advances, persistent concerns remain regarding robustness (Cai et al., 2025; Zhao et al., 2025; Huang et al., 2026) and a lack of exploration evidenced by entropy collapse (Cui et al., 2025; Shen, 2025), highlighting the need for more principled training approaches.

### 5.2 EXPLORATION IN REINFORCEMENT LEARNING

Efficient exploration remains a central challenge in Reinforcement Learning, which aim to balance between exploration and exploitation (Sutton & Barto, 2018; Weng, 2020; Amin et al., 2021). Many foundational approaches are random-based methods, such as adding Gaussian noise (Lillicrap et al., 2015), $\epsilon$-greedy policy (Sutton & Barto, 2018), or applying entropy (Haarnoja et al., 2018) or KL regularization (Vieillard et al., 2020) to encourage policy stochasticity. While simple, these methods promote randomness rather than targeted discovery, which can be suboptimal (Dann et al., 2022) and yields inconsistent performance gains in complex settings like Deep RL (Andrychowicz et al., 2021) or LLM training (Cui et al., 2025; Shen, 2025). In contrast, a more powerful paradigm directs exploration by adding an intrinsic bonus to guide agents toward novel or uncertain parts of the environment. This includes count-based approaches like UCB (Lai, 1987), LinUCB (Li et al., 2010), and LSVI-UCB (Jin et al., 2020), which use pseudo-counts of state-action visitations to explore rarely visited areas and have achieved near-optimal guarantees in bandits and linear MDPs. A parallel branch involves prediction-based methods, such as ICM (Pathak et al., 2017) and RND (Burda et al., 2018b), which use a predictive model's error as a bonus to reward the agent for reaching hard-to-predict states.

Applying these guided exploration principles is a growing field in LLM tuning. For instance, Bai et al. (2025) brings count-based principles to the RLHF process by introducing a coin flipping module. Others have focused on prediction-based bonuses. Drawing inspiration from RND, Gao et al. (2025) use an auxiliary network to predict the output of a fixed target network, with the prediction error serving as a novelty signal. Sun et al. (2025) implement an ICM-style approach, where the intrinsic reward is derived from the prediction error of a learned forward dynamics model. However, these methods require training of additional modules and rely on effective representation of responses, which is challenging (Fu et al., 2024; Barbero et al., 2024). In contrast, CDE offers a simple yet effective approach. It derives intrinsic curiosity signals directly from the actor and critics, requiring only minimal modifications to the training framework. This design yields efficient exploration that is validated both theoretically and empirically on challenging math datasets.

## 6 CONCLUSION AND FUTURE WORK

We have presented Curiosity-Driven Exploration, an efficient technique that enhances agent learning by incorporating curiosity signals from both the actor and the critic. Our approach is notably lightweight, demanding only minor modifications to the original training architecture. Its effectiveness is demonstrated by consistent accuracy improvements over strong baselines on a suite of challenging mathematical reasoning benchmarks, with these empirical results strongly corroborating our underlying theoretical framework and intuition. The **calibration collapse** revealed in our analysis aligns with recent findings on the root causes of LLM hallucination (Kalai et al., 2025), pointing to a promising avenue for future work.

## ETHICS STATEMENT

This work does not involve any ethical risks related to data privacy or misuse.

## REPRODUCIBILITY STATEMENT

To ensure reproducibility, we provide detailed descriptions of training setups and prompt used in Appendix B. The datasets used are publicly available, and we will release the code upon publication.

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

## A  THE USE OF LLMs

We utilized a Large Language Model (LLM) to proofread the manuscript for grammatical errors and to improve its clarity and readability.

## B  TRAINING DETAILS

We conduct our experiments using the `verl` training framework[2]. All models are trained with the prompt from (Yu et al., 2025b), which is illustrated in Figure 10. Detailed training configurations for CDE and the baseline models are provided in Table 6. We use a binary reward signal: $r = 1$ for a correct answer and $r = -1$ for an incorrect one. We also address a bias introduced by the training data (DAPO-17K (Yu et al., 2025b)), which consists solely of numeric answers. To prevent this from creating evaluation artifacts, we filter out problems from MATH-500 (Hendrycks et al., 2021) that require non-numeric answers.

Table 6: (a) Baseline training configurations. The **GRPO** setup is shared across all GRPO-based methods (e.g., "Qwen3-4B-Base-GRPO" and "w/PPL bonus" in Table 1); likewise, the **PPO** setup is shared across all PPO-based methods. (b) CDE-specific configurations. The PPL settings are identical for both the GRPO "w/PPL bonus" and PPO "w/PPL bonus" variants.

| Config | GRPO | PPO |
|---|---|---|
| actor-lr | 1e-6 | 1e-6 |
| critic-lr | - | 1e-5 |
| critic-warmup | - | 10 |
| kl_coef | 0.0 | 0.0 |
| max_prompt_length | 2K | 2K |
| max_response_length | 3K | 3K |
| train_batch_size | 256 | 512 |
| ppo_mini_batch_size | 256 | 256 |
| clip_ratio | - | 0.20 |
| sample temperature | 1.0 | 1.0 |
| rollout.n | 8 | 4 |
| total_training_steps | 300 | 300 |
| discount factor $\gamma$ | 1 | 1 |
| temperature | 1 | 1 |

(a)

| Config | PPL | 2,4 Heads | 8,16 Heads |
|---|---|---|---|
| $\kappa$ | 3 | 3 | 3 |
| $\alpha$ | 1 | 0.5 | 0.5 |
| $\omega$ | Staircase | No decay | No decay |
| $\zeta$ | - | 1 | 0.5 |

(b)

> Solve the following math problem step by step. The last line of your response should be of the form Answer: $Answer (without quotes) where $Answer is the answer to the problem.
> {Problem}
> Remember to put your answer on its own line after "Answer:".

Figure 10: The prompt for RLVR training.

## C  COMPUTATION COST OF MULTI-HEAD CRITIC

The additional computational overhead introduced by the multi-head critic is minimal. As illustrated in Figure 2, all heads share the same LLM backbone as the single-head critic; the only architectural difference lies in the final classification layer, whose output dimension changes from 1 to the desired number of heads. In VERL and related Transformer-based codebases, this modification corresponds simply to setting the `num_labels` argument in `AutoModelForTokenClassification.from_pretrained` to the number of heads (e.g., 4 or 16).

---

[2]https://github.com/volcengine/verl

To empirically validate that this architectural change incurs negligible overhead, we compare the memory consumption and runtime of the multi-head critic against the original single-head critic using the same Qwen3-4B-Base backbone described in the main text. Results confirm that the additional cost is marginal relative to the overall backbone computation.

Table 7: Memory Usage for Different Critic Heads

| Critic Heads | Total Parameters | Model Size (fp16) | Model Size (fp32) |
|---|---|---|---|
| **1 Head** | 4,022,470,657 | 7672.25 MB | 15344.51 MB |
| **4 Heads** | 4,022,478,340 | 7672.27 MB | 15344.54 MB |
| **16 Heads** | 4,022,509,072 | 7672.33 MB | 15344.65 MB |

To obtain a reliable measurement of computation time, we generated inputs with `batch_size=8` and `seq_len=256`, and ran both forward and backward passes of the critic for 500 iterations. We then report the averaged time consumption over these runs.

Table 8: Average Computation Time per Iteration (ms)

| Heads | Forward (ms) | Backward (ms) |
|---|---|---|
| **1** | $122.595 \pm 9.410$ | $439.958 \pm 2.787$ |
| **4** | $122.611 \pm 10.312$ | $440.713 \pm 3.036$ |
| **16** | $122.410 \pm 9.631$ | $442.260 \pm 2.425$ |

The results confirm that both memory overhead and computation time remain nearly unchanged.

## D    DETAILS ON HASH-BASED PSEUDO COUNT

The core idea is to map a full prompt–response trajectory to a compact hash code, which serves as a pseudo-state for count-based exploration. For a given trajectory, the LLM produces a sequence of last-layer hidden states, $\mathbf{h} = \{h_1, \ldots, h_{D+T}\}$, where $h_i \in \mathbb{R}^d$. Following prior work (Bai et al., 2025; Gao et al., 2025), we use the final hidden state of the last token as the trajectory's embedding, $\mathbf{h}_{traj} = h_{D+T}$.[3]

We then use SimHash (Tang et al., 2017) to generate a $k$-bit hash code. First, a fixed random projection matrix $A \in \mathbb{R}^{k \times d}$ is generated with entries drawn i.i.d. from a standard normal distribution, $\mathcal{N}(0, I)$. The hash code $\phi(q, o)$ is computed as $\phi(q, o) = \text{sign}(A\mathbf{h}_{traj}) \in \{-1, +1\}^k$. This $k$-bit vector is then mapped to an integer bucket index $b \in \{0, \ldots, 2^k - 1\}$. Let $n(b)$ be the visitation count for bucket $b$. The exploration bonus is based on this count, shaping the original reward $r(q, o)$ as follows:

$$\tilde{r}(q, o) = r(q, o) + \frac{\beta}{\sqrt{n(b)}},$$

where $\beta$ is a hyperparameter controlling the exploration bonus. This method offers lightweight, count-based guidance compared to prior work that requires matrix inversions (Li et al., 2010) or the training of additional modules (Bai et al., 2025). Our SimHash-based exploration method is detailed in Algorithm 1, with results presented in Table 9. The experiment shows that this count-based approach does not yield a significant performance increase. We hypothesize that this is due to the reasons discussed in Section 3.1, namely the difficulty of representing complex trajectories with a single embedding.

## E    ADDITIONAL EXPERIMENTAL RESULTS

---

[3]We also experimented with mean-pooling over all hidden states but found it produced less discriminative embeddings, leading to more hash collisions.

Table 9: Zero-shot accuracy of GRPO models with count based exploration bonus.

| Model | MATH Avg@1 | AMC23 Avg@16 | AMC23 Pass@16 | AIME24 Avg@16 | AIME24 Pass@16 | AIME25 Avg@16 | AIME25 Pass@16 | Avg |
|---|---|---|---|---|---|---|---|---|
| *Bonus Weight Decay Schedules* | | | | | | | | |
| Qwen3-4B-Base-GRPO | 87.3 | 63.6 | 91.1 | 21.0 | 41.9 | 20.8 | 39.2 | 48.2 |
| ↳ w Hash-bonus | 87.6 | 66.5 | 91.0 | 20.8 | 39.3 | 21.2 | 40.3 | 49.0 |

---

**Algorithm 1:** Count-based exploration for RLVR through SimHash

---

**Inputs:** Policy $\pi_\theta$; aggregator $g(\cdot)$; random projection matrix $A \in \mathbb{R}^{k \times d}$; hash counts $n[\cdot] \leftarrow 0$;
exploration coefficients $\beta$.

**for** *each training iteration* **do**

    Sample prompts $q$ and generate responses $o \sim \pi_\theta(\cdot \mid q)$

    **for** *each $(q, o)$ in the batch* **do**

        Obtain last-layer token states $\{h_i\}_{i=1}^{|q|+|o|}$

        $\mathbf{h}_{traj} \leftarrow g(\{h_i\})$         `// e.g., select the last state` $h_{|q|+|o|}$

        $\phi \leftarrow \text{sign}(A\mathbf{h}_{traj})$         `// Compute` $k$`-bit hash code`

        $b \leftarrow \text{bucket}(\phi)$         `// Convert hash code to integer index`

        $n[b] \leftarrow n[b] + 1$

        $\tilde{r}(q, o) \leftarrow r(q, o) + \beta/\sqrt{n[b]}$         `// Shape the reward with the bonus`

    Update $\pi_\theta$ using the shaped rewards $\tilde{r}(q, o)$.

---

**Analysis of Entropy Dynamics** As highlighted in prior work, entropy provides an important lens for understanding exploration ability (Cui et al., 2025), where a sharp decline in entropy often signals premature convergence and insufficient exploration. Figure 11 illustrates the entropy dynamics of baseline GRPO compared to our proposed methods. First, relative to the baseline, the PPL bonus alleviates entropy collapse, demonstrating its role in promoting exploration. Second, when comparing decay schemes, PPL with No Decay shows persistent fluctuations and fails to converge, whereas Staircase decay yields more stable entropy trajectories. This observation is consistent with our earlier findings that decaying the bonus weight is essential for ensuring stable convergence while still supporting effective exploration.

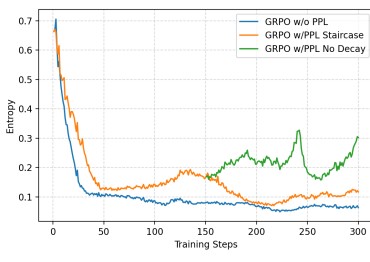

Figure 11: Dynamics of policy entropy over the training process. The bonus weight decay mechanism follows Figure 7.

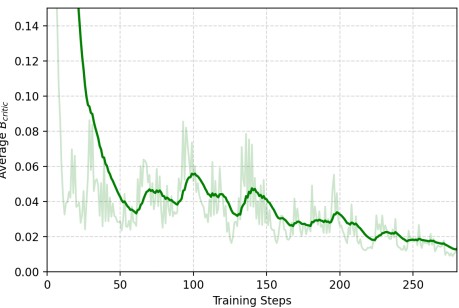

Figure 12: The average $B_{\text{critic}}$ over training steps.

**Analysis of $B_{\text{critic}}$ Dynamics** In Figure 12, we present a cross-dataset analysis by calculating the average standard deviation of the value estimates across different questions. Specifically, we evaluate three datasets: the training set (DAPO-17K), the in-domain validation set (AMC23), and the out-of-domain validation set GPQA (Rein et al., 2023). We observe that the training set exhibits a smaller standard deviation compared to both the in-domain and out-of-domain validation sets. This pattern aligns with the intuition that multi-head critics tend to show stronger disagreement on data that is less frequently encountered during training.

# F  PROOF FOR CALIBRATION THEOREM

Define $\widehat{r}_h(q, o) = r(q, o) + b_h(q, o)$ where $b_h(q, o) = \omega \min\{\kappa |r(q, o)|, -\frac{\alpha}{|o|} \log \pi_h(o|q)\}$ is a bonus function where $|o|$ is the length of response $o$. Note that $\omega$ is a redundant variable in theory because we can write $b_h(q, o) = \min\{\kappa'|r(q, o)|, -\frac{\alpha}{|o|} \log \pi_h(o|q)\}$ with $\kappa' = \omega\kappa$ and $\alpha' = \omega\alpha$. Given that $r(x, y) \in \{1, -1\}$, it suffices to consider $b_h(q, o) = \min\{\kappa, -\frac{\alpha}{|o|} \log \pi_h(o|q)\}$. Thus, as long as we use $\kappa < 1$, we have $\operatorname{sign}(\widehat{r}_h(q, o)) = \operatorname{sign}(r(q, o))$. The introduce of bonus does not change the sign of the original correctness reward.

Consider single step policy optimization

$$\pi_{h+1}(\cdot|q) = \arg\max_{\pi} \left\{ \sum_{o} \pi(o|q)\tilde{r}_h(q, o) - \frac{1}{\eta}\operatorname{KL}\left(\pi(\cdot|q)\|\pi_h(\cdot|q)\right) \right\},$$

which has closed-form solution

$$\pi_{h+1}(o|q) = \frac{\pi_h(o|q) \exp\left(\eta\widehat{r}_h(q, o)\right)}{\sum_{o'} \pi_h(o'|q) \exp\left(\eta\widehat{r}_h(q, o')\right)}.$$

For any question $q$ and response $o$. Define $Z(q) = \sum_{o'} \pi_h(o'|q) \exp\left(\eta\widehat{r}_h(q, o')\right)$, we have

$$\log \pi_{h+1}(o|q) = \log \pi_h(o|q) + \eta\widehat{r}_h(q, o) - \log\left(Z(q)\right).$$

Define $\Delta_h(o|q) = \log \pi_{h+1}(o|q) - \log \pi_t(o|q)$ as the change of likelihood of response $o$ under question $q$ at update step $t$. For two correct response $o_1^+$ and $o_2^+$ with length $|o_1^+|$ and $|o_2^+|$, and $-\frac{\alpha}{|o_1^+|} \log \pi_h(o_1^+|q) \geqslant -\frac{\alpha}{|o_2^+|} \log \pi_h(o_2^+|q)$ (i.e. $o_1^+$ has larger perplexity), we have

$$\begin{aligned}
&\Delta_h(o_1^+|q) - \Delta_h(o_2^+|q) \\
&= \widehat{r}_h(q, o_1^+) - \widehat{r}_h(q, o_2^+) \\
&= b_h(q, o_1^+) - b_h(q, o_2^+) \\
&= \min\{\kappa, -\frac{\alpha}{|o_1^+|} \log \pi_h(o_1^+|q)\} - \min\{\kappa, -\frac{\alpha}{|o_2^+|} \log \pi_h(o_2^+|q)\} \\
&\geqslant 0
\end{aligned}$$

Similarly, for two incorrect response $o_1^-$ and $o_2^-$ with $-\frac{\alpha}{|o_1^-|} \log \pi_h(o_1^-|q) \geqslant -\frac{\alpha}{|o_2^-|} \log \pi_h(o_2^-|q)$ (i.e. $o_1^-$ has larger perplexity), we have $\Delta_h(o_1^-|q) - \Delta_h(o_2^-|q) \geqslant 0$.

Specifically, given a question $q$, for any response $(o_1, o_2)$ that has the same correctness label and $-\frac{\alpha}{|o_1|} \log \pi_h(o_1|q) \geqslant -\frac{\alpha}{|o_2|} \log \pi_h(o_2|q)$, we have

- If $\widehat{r}_h(q, o_1) \geqslant \frac{1}{\eta} \log\left(Z(q)\right)$ and $\widehat{r}_h(q, o_2) \geqslant \frac{1}{\eta} \log\left(Z(q)\right)$, then $\Delta_h(o_1|q) \geqslant 0$ and $\Delta_h(o_2|q) \geqslant 0$ but $o_1$ has more likelihood increase.
- If $\widehat{r}_h(q, o_1) \geqslant \frac{1}{\eta} \log\left(Z(q)\right)$ and $\widehat{r}_h(q, o_2) < \frac{1}{\eta} \log\left(Z(q)\right)$, then $\Delta_h(o_1|q) \geqslant 0$ and $\Delta_h(o_2|q) < 0$ where $o_1$'s likelihood increase but $o_2$'s likelihood decrease.
- If $\widehat{r}_h(q, o_1) < \frac{1}{\eta} \log\left(Z(q)\right)$ and $\widehat{r}_h(q, o_2) < \frac{1}{\eta} \log\left(Z(q)\right)$, then $\Delta_h(o_1|q) < 0$ and $\Delta_h(o_2|q) < 0$ but $o_1$ has less likelihood decrease.
- It is impossible that $\widehat{r}_h(q, o_1) < \frac{1}{\eta} \log\left(Z(q)\right)$ and $\widehat{r}_h(q, o_2) \geqslant \frac{1}{\eta} \log\left(Z(q)\right)$ given that $(o_1, o_2)$ has the same correctness label and $-\frac{\alpha}{|o_1|} \log \pi_h(o_1|q) \geqslant -\frac{\alpha}{|o_2|} \log \pi_h(o_2|q)$.

# G  PROOF FOR CONSISTENCY OF MULTI-HEAD CRITIC BONUS

**Linear MDP and Assumptions**

**Assumption G.1** (Linear MDP). *We consider finite horizon $\mathcal{M} = (\mathcal{S}, \mathcal{A}, R, P, H)$ with horizon $T$, state space $\mathcal{S}$, action space $\mathcal{A}$, reward function $R : \mathcal{S} \times \mathcal{A} \to \mathbb{R}$, and transition $P : \mathcal{S} \times \mathcal{A} \to \mathcal{S}$ such that there exists a known feature $\phi \in \mathbb{R}^d$ and unknown features $\theta, \psi \in \mathbb{R}^d$ to ensure*

$$R(s, a) = \phi(s, a)^\top \theta \qquad P(s'|s, a) = \phi(s, a)^\top \psi(s').$$

*Without loss of generality, we assume $\|\phi(s, a)\| \leqslant 1$ for all $(s, a)$, and $\|\psi(s')\| \leqslant \sqrt{d}$, $\|\theta\|_2 \leqslant \sqrt{d}$.*

**Lemma G.2** (Proposition 2.3 in Jin et al. (2020)). *For linear MDPs that satisfy Assumption G.1, there exists $w_t^\star \in \mathbb{R}^d$ such that*

$$Q_t^\pi(s,a) := \mathbb{E}\Big[\sum_{h=t}^{\top} r_h \big| s_h = s, a_h = a\Big] = \phi(s,a)^\top w_t^\star.$$

The linearity of $Q$-functions enables using regression technique to solve it. Consider a dataset with $n$ observations $\mathcal{D} = \{s_{i,t}, a_{i,t}, G_{i,t}\}_{i=1}^n$ where $G_{i,t}$ is the Monte-Carlo return. Let $\phi_{i,t} = \phi(s_{i,t}, a_{i,t})$ and denote the regression noise as $\varepsilon_{i,t} = G_{i,t} - \phi_{i,t}^\top w_t^\star$. We impose the following assumptions.

(A1) $\mathbb{E}[\varepsilon_{i,t} \mid \phi_{i,t}] = 0$ and $\{(\varepsilon_{i,t})\}_{i=1}^n$ are i.i.d. $\sigma^2$ zero mean-Gaussian for each fixed $t$;

(A2) $\frac{1}{n}\sum_{i=1}^n \phi_{i,t}\phi_{i,t}^\top \xrightarrow{p} \Sigma_t > 0$

Jin et al. (2020) shows that doing value iteration on optimistically estimated Q function can achieve near-optimal regret for linear MDP, where the optimistic Q function is the combination of linear regression estimation and exploration bonus $b_{n,t} = \beta\sqrt{\phi_{n,t}^\top \Lambda_{n,t}^{-1}\phi_{n,t}}$, where $\Lambda_{n,t} = \lambda I + \sum_{i=1}^n \phi_{i,t}\phi_{i,t}^\top$ and $\beta$ is some constant. Below we will formally connect our bootstrapped bonus with this term.

### Formulation of the bootstrap multi-head critic

We accommodate the bootstrap multi-head into the linear-MDP setting. For any time step $t$, we sample $K$ mini-batches $\{S_k \subset [n]\}_{k=1}^K$ of size $m = \zeta n$ uniformly without replacement from $\mathcal{D}$ and construct the ridge estimator as follows

$$\widehat{w}_{n,t}^{(k)} = \arg\min_w \sum_{r \in S_k} (G_{r,t} - \phi_{r,t}^\top w)^2 + \zeta\lambda\|w\|^2.$$

For any feature $\phi \in \mathbb{R}^d$, we define the bootstrap multi-head bonus as

$$b_{t,K}^{\text{boot}}(\phi) = \text{std}\left(\{\phi^\top \widehat{w}_{n,t}^{(k)} | 1 \leq k \leq K\}\right).$$

**Elliptical ("count-based") bonus in (Jin et al., 2020).** The ridge estimator is constructed using all data across $n$ trajectories as follows

$$\widehat{w}_{n,t} = \arg\min_w \sum_{i=1}^n (G_{i,t} - \phi_{i,t}^\top w)^2 + \zeta\lambda\|w\|^2.$$

For any query feature $\phi \in \mathbb{R}^d$, the bonus term is $b_t^{\text{cnt}}(\phi) = \sqrt{\phi^\top \Lambda_{n,t}^{-1}\phi}$.

### Formal Version and Proof of Theorem 3.2

**Theorem G.3.** *Under Assumption G.1 and assumptions (A1)–(A2), for any fixed time-step $t$ and query $\phi \in \mathbb{R}^d$,*

$$b_{t,K}^{boot}(\phi) \xrightarrow[K\to\infty,\ n\to\infty]{p} \beta\sqrt{\phi^\top \Lambda_{n,t}^{-1}\phi},$$

*where $\beta$ is some constant.*

*Proof.* For any time-step $t$ and $S_k \subset [n]$, we have the explicit solution of the ridge regression

$$\widehat{w}_{n,t} = \Lambda_{n,t}^{-1}\sum_{i=1}^n \phi_{i,t} G_{i,t}.$$

Conditioning on $X_t = [\phi_{1,t}^\top; \ldots; \phi_{n,t}^\top]$, the conditional variance of the estimator is

$$\text{Var}\big(\phi^\top \widehat{w}_{n,t} \mid X_t\big) = \sigma^2 \phi^\top \Big(\Lambda_{n,t}^{-1} - \lambda\Lambda_{n,t}^{-2}\Big)\phi.$$

From Assumption (A2), we have

$$\lambda_{\min}(\Lambda_{n,t}) = \lambda_{\min}(\lambda I + \sum_{i=1}^n \phi_{i,t}\phi_{i,t}^\top) = \lambda + n\lambda_{\min}(\frac{1}{n}\sum_{i=1}^n \phi_{i,t}\phi_{i,t}^\top) = O_p(n). \qquad (4)$$

By definition of matrix operator norm, we have $\|\Lambda_{n,t}^{-1}\|_{\mathrm{op}} = \frac{1}{\lambda_{\min}(\Lambda_{n,t})} = O_p(1/n)$. Therefore

$$\phi^\top \Lambda_{n,t}^{-1}\phi \leqslant \|\phi\|^2 \|\Lambda_{n,t}^{-1}\|_{\mathrm{op}} = O_p(1/n) \text{ and } \phi^\top \Lambda_{n,t}^{-2}\phi \leqslant O_p(1/n^2),$$

and

$$n\phi^\top \left(\Lambda_{n,t}^{-1} - \lambda\Lambda_{n,t}^{-2}\right)\phi - n\phi^\top \Lambda_{n,t}^{-1}\phi = -n\lambda\phi^\top \Lambda_{n,t}^{-2}\phi \xrightarrow{P} 0.$$

Therefore, we have

$$\phi^\top \left(\Lambda_{n,t}^{-1} - \lambda\Lambda_{n,t}^{-2}\right)\phi \xrightarrow{P} \phi^\top \Lambda_{n,t}^{-1}\phi. \qquad (5)$$

Before moving to $b_{t,K}^{\mathrm{boot}}(\phi)$, we define the following quantities

$$\Delta\Sigma = \frac{1}{\zeta}\sum_{r\in S_k} \phi_{r,t}\phi_{r,t}^\top - \sum_{i=1}^n \phi_{i,t}\phi_{i,t}^\top, \quad b = \sum_{i=1}^n \phi_{i,t}G_{i,t}, \quad b_s = \frac{1}{\zeta}\sum_{r\in S_k} \phi_{r,t}G_{r,t}, \quad \Delta b = b_s - b.$$

Since $\Sigma_h > 0$, matrix Bernstein for sampling without replacement yields $\|\Delta\Sigma\|_{\mathrm{op}} = O_p(\sqrt{n})$. From equation 4, we have $\lambda_{\min}(\Lambda_{n,t}) \geqslant \lambda$, similarly we have $\lambda_{\min}(\Lambda_{n,t} + \Delta\Sigma) \geqslant \lambda$. Therefore, both matrices are invertible. Therefore, we have

$$(\Lambda_{n,t} + \Delta\Sigma)(\Lambda_{n,t} + \Delta\Sigma)^{-1} = I.$$

Rearranging terms yield the following form

$$(\Lambda_{n,t} + \Delta\Sigma)^{-1} = \Lambda_{n,t}^{-1} - \Lambda_{n,t}^{-1}\Delta\Sigma(\Lambda_{n,t} + \Delta\Sigma)^{-1}.$$

Substituting this expression for $(\Lambda_{n,t} + \Delta\Sigma)^{-1}$ back into the right-hand side once more yields

$$(\Lambda_{n,t} + \Delta\Sigma)^{-1} = \Lambda_{n,t}^{-1} - \Lambda_{n,t}^{-1}\Delta\Sigma\Lambda_{n,t}^{-1} + R_\Sigma,$$

where $R_\Sigma = \Lambda_{n,t}^{-1}\Delta\Sigma\Lambda_{n,t}^{-1}\Delta\Sigma(\Lambda_{n,t} + \Delta\Sigma)^{-1}$ and $\|R_\Sigma\|_{\mathrm{op}} = O_p(\|\Lambda_{n,t}^{-1}\|_{\mathrm{op}}^3\|\Delta\Sigma\|_{\mathrm{op}}^2) = O_p(1/n^2)$.

The $k$-th bootstrap ridge solution is

$$\widehat{w}_{n,t}^{(k)} = (\Lambda_{n,t} + \Delta\Sigma)^{-1}b_s.$$

Subtracting $\widehat{w}_{n,t} = \Lambda_{n,t}^{-1}b$ and inserting the expansion,

$$\widehat{w}_{n,t}^{(k)} - \widehat{w}_{n,t} = \underbrace{\Lambda_{n,t}^{-1}\Delta b - \Lambda_{n,t}^{-1}\Delta\Sigma\,\widehat{w}_{n,t}}_{\text{first order}} + \underbrace{\left(-\Lambda_{n,t}^{-1}\Delta\Sigma\Lambda_{n,t}^{-1}\Delta b + R_\Sigma b_s\right)}_{=:r_n}.$$

Since $G_{i,t} = \phi_{i,t}^\top w_t + \epsilon_{i,t}$, for any $\phi$ we have

$$\phi^\top(\widehat{w}_{n,t}^{(k)} - \widehat{w}_{n,t}) = \phi^\top\Lambda_{n,t}^{-1}\left(\frac{1}{\zeta}\sum_{r\in S_k}\phi_{r,t}\epsilon_{r,t} - \sum_{i=1}^n \phi_{i,t}\epsilon_{i,t}\right) + \phi^\top\Lambda_{n,t}^{-1}\Delta\Sigma(w_t^\star - \widehat{w}_{n,t}) + \phi^\top r_n. \quad (6)$$

Standard results for ridge regression imply that $\|w_t^\star - \widehat{w}_{n,t}\|_2 = O_p(n^{-1/2})$. Consequently, the second term is of order $O_p(n^{-1})$. Similarly, for the last term we have

$$\phi^\top r_n \leqslant \|\phi\| \left(\|\Lambda_{n,t}^{-1}\|_{\mathrm{op}}^2\|\Delta\Sigma\|_{\mathrm{op}}\|\Delta b\|_{\mathrm{op}} + \|\Delta\Sigma\|_{\mathrm{op}}\|b_s\|_{\mathrm{op}}\right) = O_p(n^{-1}).$$

We now analyze the conditional variance of the estimator $\phi^\top\widehat{w}_{n,t}^{(k)}$. Let $\mathcal{D}_n = \{(\phi_{i,t}, \epsilon_{i,t})\}_{i=1}^n$ be the fixed dataset. Conditional on $\mathcal{D}_n$, the subsampled estimator $\widehat{w}_{n,t}^{(k)}$ contains randomness solely from the index set $S_k$, while the full-sample estimator $\widehat{w}_{n,t}$ and the sum $\sum_{i=1}^n \phi_{i,t}\epsilon_{i,t}$ are constants.

Neglecting the remainder terms of order $O_p(n^{-1})$, the conditional variance is dominated by the leading term

$$\mathrm{Var}\Big(\phi^\top \widehat{w}_{n,t}^{(k)} \mid \mathcal{D}_n\Big) = \mathrm{Var}\Big(\phi^\top(\widehat{w}_{n,t}^{(k)} - \widehat{w}_{n,t}) \mid \mathcal{D}_n\Big)$$

$$\approx \phi^\top \Lambda_{n,t}^{-1} \mathrm{Var}\Big(\frac{1}{\zeta}\sum_{r\in S_k}\phi_{r,t}\epsilon_{r,t} - \sum_{i=1}^{n}\phi_{i,t}\epsilon_{i,t}\Big|\mathcal{D}_n\Big)\Lambda_{n,t}^{-1}\phi$$

$$= \frac{1}{\zeta^2}\phi^\top\Lambda_{n,t}^{-1}\mathrm{Var}\Big(\sum_{r\in S_k}\phi_{r,t}\epsilon_{r,t} \mid \mathcal{D}_n\Big)\Lambda_{n,t}^{-1}\phi.$$

Let $I_i = \mathbb{1}(i \in S_k)$ be the inclusion indicator. We can rewrite the conditional variance

$$\mathrm{Var}\Big(\sum_{r\in S_k}\phi_{r,t}\epsilon_{r,t}\Big|\mathcal{D}_n\Big) = \mathrm{Var}\Big(\sum_{i=1}^{n}I_i\phi_{i,t}\epsilon_{i,t}\Big|\mathcal{D}_n\Big).$$

From finite-population sampling theory (simple random sampling without replacement), we have the moments

$$\mathbb{E}[I_i \mid \mathcal{D}_n] = \zeta, \quad \mathrm{Var}(I_i \mid \mathcal{D}_n) = \zeta(1-\zeta), \quad \mathrm{Cov}(I_i, I_j \mid \mathcal{D}_n) = -\frac{\zeta(1-\zeta)}{n-1} \quad (i \neq j).$$

Therefore, we have

$$\frac{1}{n}\mathrm{Var}\Big(\sum_{i=1}^{n}I_i\phi_{i,t}\epsilon_{i,t}\Big|\mathcal{D}_n\Big)$$

$$= \frac{1}{n}\sum_{i=1}^{n}\mathrm{Var}(I_i \mid \mathcal{D}_n)\phi_{i,t}\phi_{i,t}^\top\epsilon_{i,t}^2 + \frac{1}{n}\sum_{i\neq j}\mathrm{Cov}(I_i, I_j \mid \mathcal{D}_n)\phi_{i,t}\phi_{i,t}^\top\epsilon_{i,t}^2$$

$$= \zeta(1-\zeta)\frac{1}{n}\sum_{i=1}^{n}\phi_{i,t}\phi_{i,t}^\top\epsilon_{i,t}^2 + \frac{\zeta(\zeta-1)}{n-1}\frac{1}{n}\sum_{i\neq j}\phi_{i,t}\phi_{j,t}^\top\epsilon_{i,t}\epsilon_{j,t}. \tag{7}$$

For the first term in equation 7, we condition on the features $\{\phi_{i,t}\}_{i=1}^{n}$. Since the noise terms $\{\epsilon_{i,t}\}_{i=1}^{n}$ are i.i.d. zero-mean Gaussian, and the features are bounded, applying the Law of Large Numbers for weighted sums yields

$$\frac{1}{n}\sum_{i=1}^{n}\phi_{i,t}\phi_{i,t}^\top\epsilon_{i,t}^2 \xrightarrow{p} \frac{1}{n}\sum_{i=1}^{n}\phi_{i,t}\phi_{i,t}^\top\sigma^2.$$

As for the second term in equation 7, we make further decomposition as follows

$$\sum_{i\neq j}\phi_{i,t}\phi_{j,t}^\top\epsilon_{i,t}\epsilon_{j,t} = (\sum_{i=1}^{n}\phi_{i,t}\epsilon_{i,t})(\sum_{i=1}^{n}\phi_{i,t}\epsilon_{i,t})^\top - \sum_{i=1}^{n}\phi_{i,t}\phi_{i,t}^\top\epsilon_{i,t}^2.$$

For the first term, we have

$$\|(\sum_{i=1}^{n}\phi_{i,t}\epsilon_{i,t})(\sum_{i=1}^{n}\phi_{i,t}\epsilon_{i,t})^\top\|_{\mathrm{op}} = \|\sum_{i=1}^{n}\phi_{i,t}\cdot\epsilon_{i,t}\|_2^2$$

Since $\mathbb{E}[\|\sum_{i=1}^{n}\phi_{i,t}\epsilon_{i,t}\|_2^2] = \sigma^2\sum_{i=1}^{n}\|\phi_{i,t}\|_2^2$, we have $\mathbb{E}[\|\sum_{i=1}^{n}\phi_{i,t}\epsilon_{i,t}\|_2^2] = O(n)$. With Markov inequality, we have $\|\sum_{i=1}^{n}\phi_{i,t}\epsilon_{i,t}\|_2^2 = O_p(n)$. As for the second term, we have $\sum_{i=1}^{n}\phi_{i,t}\phi_{i,t}^\top\epsilon_{i,t}^2 \xrightarrow{p} \sum_{i=1}^{n}\phi_{i,t}\phi_{i,t}^\top\sigma^2$ thus $\|\sum_{i=1}^{n}\phi_{i,t}\phi_{i,t}^\top\epsilon_{i,t}^2\|_{\mathrm{op}} = O_p(n)$. Therefore, for the second term in equation 7, we have $\frac{\zeta(\zeta-1)}{n-1}\frac{1}{n}\sum_{i\neq j}\phi_{i,t}\phi_{j,t}^\top\epsilon_{i,t}\epsilon_{j,t} \xrightarrow{p} 0$ and

$$\frac{1}{n}\mathrm{Var}\Big(\sum_{r\in S_k}\phi_{r,t}\epsilon_{r,t}\Big|\mathcal{D}_n\Big) \xrightarrow{p} \frac{\zeta(1-\zeta)}{n}\sum_{i=1}^{n}\phi_{i,t}\phi_{i,t}^\top\sigma^2.$$

Therefore, conditioning on $\{\phi_{i,t}\}_{i=1}^{n}$, we have

$$\mathrm{Var}\Big(\phi^\top\widehat{w}_{n,t}^{(k)} \mid \mathcal{D}_n\Big) \xrightarrow{p} \frac{\zeta-1}{\zeta}\sigma^2\phi^\top\Lambda_{n,t}^{-1}\Big(\sum_{i=1}^{n}\phi_{i,t}\phi_{i,t}^\top\Big)\Lambda_{n,t}^{-1}\phi = \frac{1-\zeta}{\zeta}\sigma^2\phi^\top\Big(\Lambda_{n,t}^{-1} - \lambda\Lambda_{n,t}^{-2}\Big)\phi$$

Then, with equation 5, we have

$$\mathrm{Var}\left(\phi^{\top}\widehat{w}_{n,t}^{(k)} \mid \mathcal{D}_n\right) \xrightarrow{p} \frac{1-\zeta}{\zeta}\sigma^2\phi^{\top}\Lambda_{n,t}^{-1}\phi.$$

Finally, by the conditional strong law of large numbers, we have

$$b_{t,K}^{\mathrm{boot}}(\phi) = \mathrm{std}\left(\left\{\phi^{\top}\widehat{w}_{n,t}^{(k)}\big|1 \leqslant k \leqslant K\right\}\right) \xrightarrow{\text{a.s.}} \sqrt{\mathrm{Var}\left(\phi^{\top}\widehat{w}_{n,t}^{(k)} \mid \mathcal{D}_n\right)} \xrightarrow{p} \sqrt{\frac{1-\zeta}{\zeta}}\sigma\sqrt{\phi^{\top}\Lambda_{n,t}^{-1}\phi}.$$

$\square$

