# OpenReview forum: "CDE: Curiosity-Driven Exploration for Efficient Reinforcement Learning in Large Language Models"
_ICLR.cc/2026/Conference — ICLR 2026 Poster_

### Official Review · Reviewer_KJQc · 2025-10-24

**Soundness:** 2
**Presentation:** 2
**Contribution:** 3
**Rating:** 6
**Confidence:** 3

**Summary:**

The paper proposes Curiosity-Driven Exploration (CDE) to improve exploration and calibration in RL-based fine-tuning of large language models. The method adds two intrinsic signals: a per-response perplexity bonus for the actor and a variance-based uncertainty bonus across bootstrapped value heads for the critic. Both are clipped and gradually reduced. Experiments on math-reasoning datasets with a Qwen-3-4B model show better accuracy and more stable confidence behavior. The contribution is mainly algorithmic and empirical, with a small theoretical part that connects the critic variance to UCB-style exploration.

**Strengths:**

- Idea that could have a large impact on the LLM community, which is easy to implement on top of existing PPO/GRPO pipelines.
- Sample-specific actor bonus helps control over-confidence and keeps diversity among correct answers.
- Critic variance is simple but works well; the UCB connection gives intuition.
- Results are consistent across multiple datasets; training curves and calibration plots are convincing.
- The paper is clear and well structured; contains ablation studies.

**Weaknesses:**

- Limited scope: all tasks are math; no larger model or other domains such as code or factual QA. How useful is it on other tasks?
- No compute/cost analysis: multi-head critics increase memory and time, but this is not measured.
- No failure cases or examples showing what happens if exploration weight is too high.
- Reproducibility: no public code/configs mentioned.
- Lacks references to traditional curious-driven approaches employed in RL community. See references for examples

@InProceedings{pmlr-v260-bougie25a,
  title     = {Exploring Beyond Curiosity Rewards: Language-Driven Exploration in RL},
  author    = {Bougie, Nicolas and Watanabe, Narimasa},
  booktitle = {Proceedings of the 16th Asian Conference on Machine Learning},
  pages     = {127--142},
  year      = {2025},
  volume    = {260},
  series    = {Proceedings of Machine Learning Research},
  month     = {December},
  publisher = {PMLR},
  url       = {https://proceedings.mlr.press/v260/bougie25a.html}
}

@inproceedings{burda2019rnd,
  title     = {Exploration by Random Network Distillation},
  author    = {Burda, Yuri and Edwards, Harrison and Pathak, Deepak and Storkey, Amos and Darrell, Trevor and Efros, Alexei A.},
  booktitle = {International Conference on Learning Representations (ICLR)},
  year      = {2019},
  url       = {https://openreview.net/forum?id=H1lJJnR5Ym}
}

@inproceedings{osband2016bootstrapped,
  title     = {Deep Exploration via Bootstrapped {DQN}},
  author    = {Osband, Ian and Blundell, Charles and Pritzel, Alexander and Van Roy, Benjamin},
  booktitle = {Advances in Neural Information Processing Systems (NeurIPS)},
  year      = {2016},
  url       = {https://proceedings.neurips.cc/paper_files/paper/2016/file/8d8818c8e140c4643106b7c59d6a9b2c-Paper.pdf}
}

**Questions:**

- Are the value targets trained with extrinsic-only rewards?
- How are the bootstrap heads decorrelated (data resampling unit, optimizer states, seeds, dropout)?
- Did you try actor-only and critic-only? Are the gains additive?
- Could you add multiple seed averages / confidence intervals on the main tables?
- Could you compare the approach with existing prior work in curiosity-driven RL?

---

> ### Author Response · Authors · 2025-11-21
> **Response to Weaknesses**
>
> We sincerely appreciate the reviewer’s recognition of our work’s contributions to the LLM community, the clarity of our manuscript, and the rigor of our experimental evaluation. These encouraging comments are invaluable for further improving our work. We address each concern in detail below.
>
> **W1: All tasks are math; no larger model or other domains.**
>
> Thank you for the helpful comment. We use math tasks because they are the standard benchmarks for recent RLVR work, and they provide a clear setting to evaluate exploration in reasoning. Importantly, CDE is a general exploration method and is not specialized to math; it should in principle apply to other domains as well.
>
> We believe our current experiments already demonstrate the effectiveness of CDE through consistent gains across multiple math datasets and ablations. To further strengthen the paper, we are running additional experiments in a different domain.
>
>
> **W2: Computation cost of multi-head critic**
>
> The additional computation introduced by the multi-head critic is marginal. As shown in Figure 2, all heads share the same LLM backbone as the single-head critic; only the final classification layer changes from output dimension 1 to the desired number of heads. In VERL and related transformers codebases, this corresponds simply to setting the num\_labels argument of AutoModelForTokenClassification.from\_pretrained to the number of heads (e.g., 4 or 16).
>
> To verify this empirically, we compared the memory usage and runtime of the multi-head critic with the original single-head critic using the same Qwen3-4B-Base backbone described in the paper.
>
> Memory:
> | Critic Heads | Total Parameters | Model Size (fp16) | Model Size (fp32) |
> | ------------ | ---------------- | ----------------- | ----------------- |
> | **1 Head**   | 4,022,470,657    | 7672.25 MB        | 15344.51 MB       |
> | **4 Heads**  | 4,022,478,340    | 7672.27 MB        | 15344.54 MB       |
> | **16 Heads** | 4,022,509,072    | 7672.33 MB        | 15344.65 MB       |
>
> To obtain a reliable measurement of computation time, we generated inputs with batch\_size = 8 and seq\_len = 256, and ran both forward and backward passes of the critic for 500 iterations. We then report the averaged time consumption over these runs.
> | Heads  | Forward (ms)         | Backward (ms)       |
> | ------ | -------------------- | ------------------- |
> | **1**  | **122.595 ± 9.410**  | **439.958 ± 2.787** |
> | **4**  | **122.611 ± 10.312** | **440.713 ± 3.036** |
> | **16** | **122.410 ± 9.631**  | **442.260 ± 2.425** |
>
> The results confirm that both memory overhead and computation time remain nearly unchanged.
>
> **W3: Failure cases showing when exploration weight is too high**
>
> Thank you for the suggestion. Our ablation on the weight-decay mechanism already demonstrates the failure mode when the exploration weight remains high during the training. As shown in the figure, the entropy remains abnormally high throughout training, and the model fails to improve over the baseline, which an indication of over-exploration and poor convergence when the exploration weight is too high.
>
> We also plan to conduct additional sensitivity analyses on other hyperparameters, which may illustrate other potential failure cases.
>
> **W4: Reproducibility**
>
> Appendix B provides detailed hyperparameter settings, training configurations, and all prompts used during training. We believe these materials are sufficient for reproducing our results. In addition, we will release the full codebase upon acceptance to ensure complete reproducibility.
>
>
> **W5: Reference to curiosity driven RL**
>
> We have already cited Bootstrapped DQN[1] when discussing the multi-head critic (line 243), and we reference several curiosity-driven exploration methods in the RL literature, including RND[2] and others in Section 5. Regarding the suggested reference (Bougie et al., 2025)[3], it uses LLMs to guide the training of RL policies, which differs from our setting of training LLMs themselves, and is therefore less directly related. That said, we are happy to include it in the related-work section to provide broader context for readers.

---

> ### Author Response · Authors · 2025-11-21
> **Response to Questions**
>
> **Q1: Only extrinsic reward used to train the critic?**
>
> Yes. As stated around line 294, we train the critic solely using the extrinsic reward signal (±1).
>
> **Q2: How to de-correlate the heads?**
>
> First, all value heads are initialized differently, which already provides an initial source of de-correlation. Prior work, most notably Bootstrapped DQN [1], has explicitly observed that best performance is attained simply relying on different initializations (i.e., using a full-sharing mask $\zeta = 1$). Our own ablations on $\zeta$ corroborate this phenomenon: in several settings, simply using $\zeta = 1$ already leads to sufficient diversity for exploration.
>
> In experiments with masking fractions $\zeta < 1$, each head is further updated on a distinct subset of the data, introducing an additional source of stochasticity and strengthening the independence among heads.
>
> **Q3: Only actor/critic? Is the benefit addable?**
>
> In Table 2, we report results using only the actor curiosity bonus or only the critic curiosity bonus. The two signals are complementary, but we find that exploration improvements tend to saturate once one signal is sufficiently strong. In practice, our experiments combining the multi-head bootstrap bonus with the actor bonus show similarly limited additional gains.
>
> | **Model**                    | **MATH** Avg@1 | **AMC23** Avg@16 | **AMC23** Pass@16 | **AIME24** Avg@16 | **AIME24** Pass@16 | **AIME25** Avg@16 | **AIME25** Pass@16 | **Avg** |
> | ---------------------------- | -------------- | ---------------- | ----------------- | ----------------- | ------------------ | ----------------- | ------------------ | ------- |
> | Qwen3-4B-Base-PPO            | 86.6           | 64.1             | 87.2              | 17.8              | 36.0               | 17.5              | 33.7               | 46.5    |
> | → **w/ 4 Heads**             | 87.3           | 63.9             | 87.9              | 21.5              | 35.5               | 21.5              | 45.5               | 48.5    |
> | → **w/ 4 Heads & PPL bonus** | 86.3           | 65.6             | 88.0              | 20.8              | 38.5               | 21.3              | 33.5               | 48.5    |
>
> **Q4: Add confidence intervals in the table?**
>
> We appreciate the reviewer’s suggestion to include statistics in the main tables. We have updated the results to report the standard deviation of the Avg@16 evaluation directly in the main table. Please refer to the revised PDF for the updated numbers.
>
> **Q5: Compare to other curiosity baselines?**
>
> Thank you for the suggestion regarding comparisons to other curiosity-driven methods. To our knowledge, only one curiosity driven method for RLVR that provides publicly available code [4]. This work is a very recent arXiv preprint whose code was released in July, so per the conference review policy, a direct comparison was not required.
>
> That said, we agree that including this baseline adds valuable context and strengthens our submission. We have therefore conducted additional experiments with i-MENTOR and report the results below. As shown, i-MENTOR-GRPO provides a modest improvement over the GRPO baseline, whereas i-MENTOR-PPO fails to improve upon PPO. In contrast, our method outperforms i-MENTOR on both GRPO- and PPO-based training, further demonstrating the effectiveness and robustness of our approach.
>
> | **Model**                 | MATH Avg@1 | AMC23 Avg@16 | AMC23 Pass@16 | AIME24 Avg@16 | AIME24 Pass@16 | AIME25 Avg@16 | AIME25 Pass@16 | **Avg** |
> | ------------------------- | ---------- | ------------ | ------------- | ------------- | -------------- | ------------- | -------------- | ------- |
> | Qwen3-4B-Base-GRPO        | 87.3       | 63.6         | 89.1          | 20.8          | 41.9           | 21.0          | 39.2           | 48.2    |
> | → **i-MENTOR-GRPO**       | 87.6       | 63.2         | 89.1          | 22.5          | 39.3           | 23.0          | 40.4           | 49.1    |
> | → **w/ PPL bonus (Ours)** | 87.7       | 67.8         | 89.5          | 23.3          | 48.5           | 23.5          | 42.5           | 50.6    |
> | Qwen3-4B-Base-PPO         | 86.6       | 64.1         | 87.2          | 17.8          | 36.0           | 17.5          | 33.7           | 46.5    |
> | → **i-MENTOR-PPO**        | 85.8       | 60.9         | 85.9          | 18.9          | 39.4           | 19.6          | 36.5           | 46.3    |
> | → **w/ 4 Heads (Ours)**   | 87.3       | 63.9         | 87.9          | 21.5          | 35.5           | 21.5          | 45.5           | 48.5    |

---

> ### Author Response · Authors · 2025-11-21
> **References**
>
> **References**
>
> [1] Osband, Ian, et al. "Deep exploration via bootstrapped DQN." Advances in neural information processing systems 29 (2016).
>
> [2] Burda, Yuri, et al. "Exploration by random network distillation." International Conference on Learning Representations.
>
> [3] Bougie, Nicolas, and Narimasa Watanabe. "Exploring Beyond Curiosity Rewards: Language-Driven Exploration in RL." The 16th Asian Conference on Machine Learning (Conference Track). 2024.
>
> [2] Gao, Jingtong, et al. "Navigate the unknown: Enhancing llm reasoning with intrinsic motivation guided exploration." arXiv preprint arXiv:2505.17621 (2025).

---

> > ### Comment · Reviewer_KJQc · 2025-11-21
> >
> > I thank the authors for their thorough comments and for answering my questions as well as adding new experiments.
> > I will maintain my current score.

---

### Official Review · Reviewer_XET9 · 2025-10-26

**Soundness:** 3
**Presentation:** 4
**Contribution:** 4
**Rating:** 6
**Confidence:** 3

**Summary:**

This paper introduces 'Curiosity-Driven Exploration' (CDE) to improve RLVR exploration. It proposes two bonuses: perplexity (PPL) for GRPO and multi-head critic variance for PPO. The method is shown to fix a "calibration collapse" where baselines become overconfident in their errors.

**Strengths:**

* The proposed PPL bonus is a simple, intuitive, and well-motivated method for exploration. The paper provides compelling evidence that this simple bonus directly fixes the identified calibration collapse.
* The paper is well-written and easy to follow.

**Weaknesses:**

* The experiment setting is a little vague. It is not clear in Table 1 whether the experiments on PPO with description `w/ x Heads` are combined with PPL.
    * If yes, why `w/ 2 Heads` is worse than `w/ PPL`?
    * If not, had the authors tried to combine PPL with a multi-head critic, and what are the results?
* The discovery that a "Staircase" decay (i.e., a hard stop) for the PPL bonus is optimal (Table 2) is counterintuitive and feels ad hoc. The paper would be stronger if it investigated why a hard cutoff is so much better than a smooth anneal.
* The paper's argument for dismissing count-based exploration seems contradictory. Section 3.1 argues that hash-based counting fails due to the "poor expressiveness" of LLM embeddings. However, the proposed multi-head critic method (Section 3.3) fundamentally relies on these same LLM hidden-state representations to feed its value heads. The paper does not resolve why these representations are considered too poor for hashing but sufficient for learning complex value functions.

**Questions:**

Please refer to the weaknesses.

---

> ### Author Response · Authors · 2025-11-21
> **Response to Weaknesses**
>
> We sincerely appreciate the reviewer’s recognition of the intuitiveness of our approach, the clarity of the manuscript, and the strength of the presented evidence. These encouraging comments are invaluable to us. We address each concern in detail below.
>
> **W1: Is multi-head combined with PPL?**
>
> Thanks for pointing this out. The w/ x Heads results in Table 1 is not combined with PPL. Yes, we have tried to combine PPL with a multi-head critic. The two signals seem complementary, but we find that exploration improvements tend to saturate once one signal is sufficiently strong. In practice, our experiments combining the multi-head bootstrap bonus with the actor bonus show similarly limited additional gains.
>
> | **Model**                    | **MATH** Avg@1 | **AMC23** Avg@16 | **AMC23** Pass@16 | **AIME24** Avg@16 | **AIME24** Pass@16 | **AIME25** Avg@16 | **AIME25** Pass@16 | **Avg** |
> | ---------------------------- | -------------- | ---------------- | ----------------- | ----------------- | ------------------ | ----------------- | ------------------ | ------- |
> | Qwen3-4B-Base-PPO            | 86.6           | 64.1             | 87.2              | 17.8              | 36.0               | 17.5              | 33.7               | 46.5    |
> | → **w/ 4 Heads**             | 87.3           | 63.9             | 87.9              | 21.5              | 35.5               | 21.5              | 45.5               | 48.5    |
> | → **w/ 4 Heads & PPL bonus** | 86.3           | 65.6             | 88.0              | 20.8              | 38.5               | 21.3              | 33.5               | 48.5    |
>
> **W2: Explanation to Staircase decay**
>
> We agree the superiority of the "staircase" decay over a smooth anneal is not immediately obvious. As we discussed in Section 4.3 (L364-L377), our interpretation is that the model's performance is sensitive to the "exploration budget" received during the early-learning phase.
>
> As shown in Figure 7, the staircase schedule provides the largest area under the curve for the bonus weight during the initial phase. In contrast, linear and cosine schedules begin to "starve" the model of this exploration bonus much earlier. The strong performance of the staircase schedule in Table 2 empirically confirms this; models trained with linear or cosine decays underperform because the bonus is reduced too quickly, prematurely weakens the exploration stage. We will add more discussion on this phenomenon in the revision.
>
>
> **W3: Contradictory in dismissing count-based exploration**
>
> First, we would like to clarify that the actor and critic are two separate LLMs, and they do not share parameters or hidden-state representations.
>
> Our analysis in Section 3.1 specifically concerns the actor’s hidden states, which are optimized solely for next-token generation. Such representations are not trained to distinguish long reasoning trajectories, and one recent work [1] has shown that last-layer hidden states have the ``over-squashing'' issue, meaning they tend to encode only coarse semantic information rather than detailed structural differences. Because count-based exploration relies on the embedding space to differentiate reasoning trajectories with high sensitivity, the actor’s representation is insufficiently expressive for hashing-based pseudo-counts, leading to the failures we report.
>
> This does not contradict the effectiveness of our proposed multi-head critic. In the critic, the hidden states are learned end-to-end together with the value heads, and are explicitly optimized for accurate value prediction. Just as the actor’s hidden states support high-quality next-token modeling, the critic’s hidden states become well-suited for value estimation. Consequently, disagreement among the value heads provides a meaningful exploration signal.
>
> **References**
>
> [1] Barbero, Federico, et al. "Transformers need glasses! information over-squashing in language tasks." Advances in Neural Information Processing Systems 37 (2024): 98111-98142.

---

> > ### Comment · Reviewer_XET9 · 2025-11-21
> >
> > Thank the authors for the clarifications. I will keep my current score, which leans towards accepting this paper.

---

### Official Review · Reviewer_fSmQ · 2025-10-29

**Soundness:** 3
**Presentation:** 3
**Contribution:** 2
**Rating:** 6
**Confidence:** 3

**Summary:**

The paper first conducts preliminary experiments showing that counting-based exploration methods using response embeddings are ineffective for LLM RL and then proposes Curiosity-Driven Exploration (CDE) to guide exploration during LLM RLVR.
The framework contains two parts:
   - exploration guided by actor curiosity: uses the negative seq logprob for reward shaping, with clipping and scaling coefficient to avoid reward hacking.
   - exploration guided by critic curiosity: uses a critic model with multiple heads, each trained on a subset of data, and uses the ensemble value for GAE while taking the standard deviation across heads as an exploration bonus.

Experiments mainly on Qwen3-4B-Base and math demonstrate the effectiveness of CDE, and additional experiments show that actor curiosity (negative logprob) can improve calibration.

**Strengths:**

- The paper is easy to follow, with clear motivation and structure. The SimHash attempt is also informative.
- Experiments show nontrivial improvements over vanilla GRPO and PPO baselines, supported by ablations on specific hyperparameters.
- The calibration analysis is also a bonus.

**Weaknesses:**

- Using the negative log-probability for reward shaping is not entirely new. The same formulation has been explored in related work [1], and it is also closely related to entropy shaping, which is also widely explored in related works [2, 3]. The clipping mechanism involving $\lvert A \rvert / \kappa$ is also conceptually similar to that in [3]. While direct comparisons are not strictly required by policy, discussing these related approaches would help position the contribution of this paper more clearly.
- The proposed framework introduces several new hyperparameters, yet only the ablations for coefficient scheduling for GRPO and sub-sampling fraction for critic update are provided. Given the number of hyperparameters, it would be worthwhile to conduct a more comprehensive sensitivity analysis.
- All experiments are conducted on Qwen3-4B-Base and math datasets. While I understand that it might be computationally expensive to scale up, it would be worthwhile to at least conduct experiments on other model families. To be honest, I won't be fully convinced by "Qwen + math" combination. Also considering the current field with various GRPO variants, in my opinion, vanilla GRPO is a weak baseline.

[1] Know When to Explore: Difficulty-Aware Certainty as a Guide for LLM Reinforcement Learning, arxiv 2025.

[2] GTPO and GRPO-S: Token and Sequence-Level Reward Shaping with Policy Entropy, arxiv, 2025.

[3] Reasoning with Exploration: An Entropy Perspective on Reinforcement Learning for LLMs, arxiv 2025.

**Questions:**

Please refer to the weakness section for main questions.
 - I am a bit confused about the training setup in this paper. Are you doing on-policy or off-policy training? The entropy curve of GRPO w/o PPL in Figure 11 looks somewhat abnormal to me, I would expect a higher entropy level if using clip-higher in off-policy setting. And clip-higher should be a stronger baseline. So I checked the hyperparameters table, and based on the bs and mini_bs, it appears to be on-policy, but clip_ratio is also listed, which typicality has no effect in on-policy training. Could you clarify the exact training setting used in these experiments? If it is indeed on-policy, have you also validated your methods under an off-policy setting?
- The footnote explaining PPL bonus should probably be moved to the first page, since you start using this notation in the introduction, but the clarification appears a little late.

---

> ### Author Response · Authors · 2025-11-20
> **Response to Weaknesses**
>
> We sincerely appreciate your recognition of the value of our work, the intuitiveness of the SimHash experiment, and the rigor of our experimental evaluation. We address each concern below.
>
> **W1: Discussion with related works.**
>
> We appreciate the reviewer’s constructive feedback and clarification regarding the ICLR policy. We are on the same page that comparisons to contemporaneous work are not mandatory, especially for papers [1] and [2] which appeared on arXiv in late August 2026, concurrent with our submission period. Nevertheless, we acknowledge that clarifying the relationship to these recent developments can help further contextualize our contribution.
>
> Relation to DACE [1].
>
> DACE also employs negative log-probability as a shaping signal, similar in spirit to the actor curiosity bonus in CDE. However, the two methods mainly differ in different ways. First, both works share the principle of preserving the integrity of the labeled reward. CDE uses truncation while DACE uses a small scaling factor. Secondly, both works consider the exploration-exploitation trade-off. CDE uses a decayed exploration bonus while DACE by forcing easy tasks to exploit while hard tasks to explore. Generally speaking, DACE serves as a good complementary of CDE. Combining problem difficulty with bonus assignment may serve as an interesting future research direction.
>
> Relation to GTPO / GRPO-S [2] and [3].
>
> Both works fall under entropy-based reward or advantage shaping. The core idea of GTPO is to encourage exploration for correct responses while penalizing overly confident mistakes—conceptually aligned with Figure 3 and Theorem 3.1 in our paper.  However, as discussed on Page 5, lines 216–227, our actor curiosity bonus can be seen as a sample-specific refinement of entropy shaping, offering a more targeted exploration signal. [3] incorporates an entropy clipping mechanism, which is consistent with the clipping strategy used in CDE, further supporting the design choices in our formulation.
>
> We thank the reviewer again for pointing us to these connections. To better assist readers, we will add them into related works.
>
> **W2: More comprehensive sensitivity analysis.**
>
> Thank you for raising this point. We agree that sensitivity analysis is important. We have already conducted several meaningful sensitivity studies, including those on the coefficient scheduling for GRPO and the critic subsampling fraction, which we believe to be the most critical hyperparameters to examine. We will also prioritize additional sensitivity analysis and provide further analysis on other parameters by evaluating different combinations of ($\kappa$) and ($\alpha$), offering a more comprehensive picture.
>
> **W3: Other model, dataset and benchmark.**
>
> Thank you for the helpful comment. We use math tasks because they are the standard benchmarks for recent RLVR work, and they provide a clear setting to evaluate exploration in reasoning. Importantly, CDE is a general exploration method and is not specialized to math; it should in principle apply to other domains as well.
>
> We believe our current experiments already demonstrate the effectiveness of CDE through consistent gains across multiple math datasets and ablations. To further strengthen the paper, we are running additional experiments in a different domain with a different model family.
>
> As for other GRPO variants, we have conducted experiment with i-MENTOR-GRPO and i-MENTOR-PPO [1], another curiosity driven exploration variant of GRPO and PPO. The experimental results are as follows:
>
> | **Model**                 | MATH Avg@1 | AMC23 Avg@16 | AMC23 Pass@16 | AIME24 Avg@16 | AIME24 Pass@16 | AIME25 Avg@16 | AIME25 Pass@16 | **Avg** |
> | ------------------------- | ---------- | ------------ | ------------- | ------------- | -------------- | ------------- | -------------- | ------- |
> | Qwen3-4B-Base-GRPO        | 87.3       | 63.6         | 89.1          | 20.8          | 41.9           | 21.0          | 39.2           | 48.2    |
> | → **i-MENTOR-GRPO**       | 87.6       | 63.2         | 89.1          | 22.5          | 39.3           | 23.0          | 40.4           | 49.1    |
> | → **w/ PPL bonus (Ours)** | 87.7       | 67.8         | 89.5          | 23.3          | 48.5           | 23.5          | 42.5           | 50.6    |
> | Qwen3-4B-Base-PPO         | 86.6       | 64.1         | 87.2          | 17.8          | 36.0           | 17.5          | 33.7           | 46.5    |
> | → **i-MENTOR-PPO**        | 85.8       | 60.9         | 85.9          | 18.9          | 39.4           | 19.6          | 36.5           | 46.3    |
> | → **w/ 4 Heads (Ours)**   | 87.3       | 63.9         | 87.9          | 21.5          | 35.5           | 21.5          | 45.5           | 48.5    |

---

> ### Author Response · Authors · 2025-11-21
> **Response to Questions**
>
> **Q1: About on/off-policy.**
>
> Your observation is insightful. We indeed adopt an on-policy training setup. We haven't validated GRPO methods under an off-policy settings yet.
>
> Regarding the reviewer's intuition that "clip-higher" should serve as a stronger baseline or yield higher entropy, we respectfully point to recent literature that suggests otherwise. [2] systematically examined the effect of clip-higher and found that "for the base models, adjusting the upper clipping value yields minor effects on policy entropy and even damages the performance compared to the vanilla policy" (see Figures 8 and 9 in [2]). They find out the reason behind such ineffectiveness is "the base models operate with a low policy clipping rate, approximately 0.003, which indicates only minimal deviation between successive policies". Also, we want to point out the entropy curve in [2] share a similar trend as our Figure 11, with or without clipping-higher.
>
> To rigorously address the reviewer’s concerns, we conducted a "light-weight" experiment training two off-policy version of the GRPO baseline (mini-batch size = 128), keeping other parameters the same. One with clip-higher (clip-ratio-high=0.28) and one without. We call them GRPO-high, GRPO-base. We run for a 50 steps and have following findings:
>
> 1. The entropy of both GRPO-high and GRPO-base drop down to around 0.1 after 40 steps, similar to that of Figure 11.
>
> 2. The policy clip-fraction of GRPO-base is around 0.005, with GRPO-high around 0.003. Validating the findings in [2] that clipping rate is small for base-models, so that clip-higher offers less impact on entropy and model performance.
>
> We hope these explanations and the experiments can address the reviewer's concerns. While we acknowledge the value of a full-scale off-policy study, given the limited rebuttal time-frame, we have prioritized completing the experiments requested in W2 and W3, as they are critical to demonstrating the core effectiveness of our proposed CDE method.
>
> **Q2: About footnote.**
>
> Thanks reviewer for the good advice, we have moved the footnote to the first page.
>
> **References**
>
> [1] Li, Ang, et al. "Know when to explore: Difficulty-aware certainty as a guide for llm reinforcement learning." arXiv preprint arXiv:2509.00125 (2025).
>
> [2] Tan, Hongze, et al. "Gtpo and grpo-s: Token and sequence-level reward shaping with policy entropy." arXiv preprint arXiv:2508.04349 (2025).
>
> [3] Cheng, Daixuan, et al. "Reasoning with exploration: An entropy perspective." arXiv preprint arXiv:2506.14758 (2025).
>
> [4] Gao, Jingtong, et al. "Navigate the unknown: Enhancing llm reasoning with intrinsic motivation guided exploration." arXiv preprint arXiv:2505.17621 (2025).
>
> [5] Liu, Zihe, et al. "Part i: Tricks or traps? a deep dive into rl for llm reasoning." arXiv preprint arXiv:2508.08221 (2025).

---

> > ### Comment · Reviewer_fSmQ · 2025-11-25
> >
> > Thank the authors for the clarification and additional experiments. While I would prefer to see experiments extended to other model families, I will maintain the scores.

---

> > > ### Author Response · Authors · 2025-12-01
> > >
> > > Hope you had a wonderful Thanksgiving break.
> > >
> > > Following up on your earlier comments, we have now completed the additional CDE experiments on **Llama-3.2-3B-Instruct** using **DAPO-17K**, with all other settings kept identical. This provides evidence from a different model family. The results show that **CDE outperforms GRPO and PPO by approximately +2 and +4 points**, respectively. We hope these new findings further strengthen the manuscript, and we would be grateful if the reviewer could consider rasing the score accordingly.
> > >
> > > | **Model**                  | MATH Avg@1 | AMC23 Avg@16 | AMC23 Pass@16 | AIME24 Avg@16 | AIME24 Pass@16 | **Avg** |
> > > | -------------------------- | ---------- | ------------ | ------------- | ------------- | -------------- | ------- |
> > > | Llama-3.2-3B-Instruct-GRPO | 53.8       | 30.0         | 59.4          | 13.1          | 30.7           | 32.3    |
> > > | → **w/ PPL bonus (Ours)**  | 55.8       | 32.8         | 65.0          | 14.6          | 35.1           | 34.4    |
> > > | Llama-3.2-3B-Instruct-PPO  | 51.9       | 27.2         | 59.1          | 12.5          | 29.2           | 30.5    |
> > > | → **w/ 4 Heads (Ours)**    | 56.1       | 32.2         | 63.7          | 14.3          | 35.5           | 34.2    |
> > >
> > > Additionally, we conducted a more comprehensive sensitivity analysis:
> > >
> > > | **Model**                                 | **MATH** Avg@1 | **AMC23** Avg@16 | **AMC23** Pass@16 | **AIME24** Avg@16 | **AIME24** Pass@16 | **AIME25** Avg@16 | **AIME25** Pass@16 | **Avg** |
> > > | ----------------------------------------- | -------------- | ---------------- | ----------------- | ----------------- | ------------------ | ----------------- | ------------------ | ------- |
> > > | Qwen3-4B-Base-PPO                         | 86.6           | 64.1             | 87.2              | 17.8              | 36.0               | 17.5              | 33.7               | 46.5    |
> > > | → **$\kappa=3 $,$\alpha=0.5$** (Original) | 87.3           | 63.9             | 87.9              | 21.5              | 35.5               | 21.5              | 45.5               | 48.5    |
> > > | → **$\kappa=1$,$\alpha=1$**               | 86.5           | 65.0             | 89.5              | 20.8              | 42.5               | 20.5              | 39.8               | 48.2    |
> > >
> > > | **Model**                               | **MATH** Avg@1 | **AMC23** Avg@16 | **AMC23** Pass@16 | **AIME24** Avg@16 | **AIME24** Pass@16 | **AIME25** Avg@16 | **AIME25** Pass@16 | **Avg** |
> > > | --------------------------------------- | -------------- | ---------------- | ----------------- | ----------------- | ------------------ | ----------------- | ------------------ | ------- |
> > > | Qwen3-4B-Base-GRPO                      | 87.3           | 63.6             | 89.1              | 20.8              | 41.9               | 21.0              | 39.2               | 48.2    |
> > > | → **$\kappa=3 $,$\alpha=1$** (Original) | 87.7           | 67.8             | 89.5              | 23.3              | 48.5               | 23.5              | 42.5               | 50.6    |
> > > | → **$\kappa=1$,$\alpha=1$**             | 20.9           | 11.1             | 54.6              | 1.5               | 12.7               | 1.3               | 11.4               | 8.67    |
> > > | → **$\kappa=2$,$\alpha=1$**             | 87.6           | 66.5             | 86.9              | 22.3              | 42.5               | 21.6              | 40.1               | 49.5    |
> > > | → **$\kappa=4$,$\alpha=1$**             | 88.6           | 68.2             | 90.1              | 23.6              | 46.6               | 22.7              | 40.8               | 50.8    |
> > >
> > > The critic-wise bonus is generally robust to hyperparameter choices; setting both $\kappa$ and $\alpha$ to 1 produces nearly identical performance. The actor-wise bonus is relatively more sensitive. Generally, to get realative good performance, the clipping fraction $\kappa$ should not be set to small, i.e. the bonus should be constrained to be not exceeding at least 50/% of the labeled reward.

---

### Official Review · Reviewer_XcFv · 2025-11-01

**Soundness:** 1
**Presentation:** 2
**Contribution:** 1
**Rating:** 0
**Confidence:** 4

**Summary:**

+ Summary & Contributions
	- The authors focus their attention on the exploration challenge in RL fine-tuning of LLMs.
	- Following suit with curiosity-driven exploration strategies in deep RL, the authors propose their own curiosity-driven exploration methods for RL fine-tuning.
	- Following an actor-critic setup for RL fine-tuning, the authors propose using the perplexity of the actor's response distribution as an intrinsic reward for guiding exploration. Meanwhile, the authors employ a critic ensemble with bootstrap sampling to approximate the true Bayesian posterior over the value function induced by the current actor policy and use the corresponding ensemble variance as an intrinsic reward to further guide exploration.
	- A theoretical result is provided to show that the use of the ensemble critic standard deviation to guide exploration is an asymptotically consistent estimator of the elliptical potential bonus used by LSVI-UCB (a statistically-efficient exploration algorithm) under a linear MDP assumption.
	- Empirical results show that the proposed curiosity-based reward for exploration yield improved performance relative to standard GRPO and PPO.

**Strengths:**

+ Quality
	- Strengths
		- The authors display a nice insight in trying to leverage ideas for exploration strategies from deep RL to improve LLM exploration during RL fine-tuning.
	- Weaknesses
		* Major
			- There is absolutely no scientific evidence offered in this paper to support the assertion that LLM reasoning aligns with models of childhood exploration in developmental psychology (L69-72, L151-155). This kind of hyperbole is also rather pointless as the authors could just introduce their method without unsubtantiated nonsense to give their method the veneer of being grounded in cognitive science. As the authors already point out, there is much deep RL work leaning on ideas of curiosity which already serves as a sufficient foundation to warrant their study in the context of LLMs and RL-based fine-tuning.
			- In light of the authors' proposed curiosity signals as intrinsic rewards, they proceed to augment the computation of advantages (L273). This runs counter to how intrinsic motivation is normally applied in (deep) RL, where the underlying policy optimization algorithm (PPO, GRPO, etc.) remains entirely unchanged and is simply run on the augmented MDP whose reward function has been modified to include the original environment reward plus a (potentially) weighted intrinsic reward term. Why did the authors feel compelled to deviate from this recipe? Why is it necessary or even sensible to modify the definition of the advantage function based on curiosity?
			- In motivating their proposed actor curiosity signal based on surprisal (not perplexity -- see minor comment below), the authors claim that a response with low probability under the current policy "resides in an underexplored region of its learned distribution." An alternative but equally (if not more) plausible possibility is that this response, either during the initial pre-training stage or in earlier rounds of fine-tuning, was determined to not yield correct responses, lead to negative advantages, and has accordingly had its policy of being generated reduce through successive policy-gradient updates. How are the authors able to take this surprisal curiosity signal and disambiguate between inexperience with a particular response from the implausibility of that response being optimal/correct? The fact that they can't seem to distinguish between those two critical scenarios is perhaps what warrants the subsequent clipping scheme (Equation 2).
			- The proof of Theorem 3.2 isn't actually a complete proof. There are numerous points where the authors simply claim a particular equation or inequality without any kind of explicit, step-by-step justification. Where are the steps to establish how Assumption A2 leads to $||\Lambda^{-1}_{n,t}||_{\text{op}} = O_p(\frac{1}{n})$? Where did the expansion in L895 come from? It seems like a consequence of the Sherman-Morrison-Woodbury matrix identity, but the whole point is to give a complete proof detailing those steps. Equation 4 is just stated as a matter of fact without any concrete steps as is the subsequent inequality. The authors vaguely gesture towards "finite-population sampling theory" without so much as a single citation of an existing result being utilized to then state an equation for "Var*" (which denotes some entirely unspecified variance -- why *?). Overall, the proof is entirely incomplete and, more worryingly, seems to be an amalgamation of equations and inequalities likely taken from another proper RL theory paper (the LSVI-UCB paper perhaps) just to inject math into the paper for the sake of justifying the authors heuristic approach (see "Mathiness" in [2]).
		* Minor
			- What the authors continuously refer to as the perplexity of the actor is, in reality, just the surprisal [1]. The authors even acknowledge that what they refer to isn't actually the perplexity in a footnote (L215). What is the point of erroneously labeling the surprisal as the perplexity?
			- While being able to give names to each of the injected hyperparameters in Equation 2 seems useful, it results in an unnecessarily convoluted equation. Either there is an $\frac{\omega}{\kappa}$ term which can be consolidated into a single hyperparameter or a $\omega \cdot \alpha$ term which can also be consolidated. Why can't Equation 2 follow the standard form of $r(q,o) + \lambda \cdot B_{\text{actor}}(q,o)$?
			- The authors claim that incorrect responses recieve a "larger penalty" by virtue of a smaller PPL bonus. Notice that a larger penalty is not the same thing as smaller reward bonus; relative to other actions, this might still end up inducing a positive advantage an increasing probability mass for incorrect responses. Also, it seems like the convolute structure of Equation 2 means this could be undone anyways through the $\kappa$ parameter?
			- Calling Theorem 3.1 a theorem is rather generous. The authors have stepped through the possible cases for the signs of log-probability differences.
			- The authors mention a "posterior distribution" when discussing actor-critic methods. What is this posterior distribution over? Where did it come from? Actor-critic methods do not engage with any kind of Bayesian posterior distribution by default.
			- There is some exposition concluding Section 3.2 where the authors distinguish their incorporation of a surprisal exploration bonus as distinct from traditional entropy regularization. I agree with this, however what they do end up convincing me of is that their proposed surprisal bonus may be recovering (either entirely or partially) the objective tackled by maximum entropy (maxent) RL approaches. In this case, a more appropriate baseline to compare to would be Soft Actor Critic (SAC) [3], rather than PPO.
			- It's not clear why averaging is the correct thing to do for deriving an alternative GAE for the ensemble critic. If each ensemble member represents a distinct hypothesis about the underlying $V^\pi$ then its not clear why averaging would be sensible. If anything, a posterior distribution over $V^\pi$ implies a posterior over $Q^\pi$ and, consequently, the advantage function itself.

+ Clarity
	- Strengths
		- Overall, the paper is reasonably written and reasonably structured.
	- Weaknesses
		* Major
			- N/A
		* Minor
			- The presented results for a hasing based technique seem rather out of place for this work. At the very least, it seems like the space could be better utilized for some other purpose more central to the main contributions of this work and the presented result could be relegated to the appendix with little impact on the paper.


+ Originality
	- Strengths
		- The incorporation of the specific proposed curiosity signals to enhance exploration in RL fine-tuning of LLMs is, to the best of my knowledge, novel.
	- Weaknesses
		* Major
			- It would be surprising to me if the authors are in fact the first to propose intrinsic motivation as a remedy for the well-known entropy loss/response distribution collapse issues known to plague RL fine-tuning of LLMs. Have the authors done a proper literature review for other intrinsic rewards aimed at remedying the same problem. Some of these seem to have been identified in Section 5 and yet not compared against as baselines in the reported experiments.
			- While curiosity-based exploration and intrinsic motivation are two remedies for facilitating better exploration in RL, they are by no means the only mechanisms for doing so. Indeed, existing work has already explored other avenues for addressing the same exploration challenge using different techniques [4,5]. The authors have done the bare minimum in their experiments, offering standard PPO/GRPO as baseline comparisons. A more thorough empirical evaluation would properly compare against other competing, alternative exploration strategies that go outside the avenue of curiosity/intrinsic motivation advocated for in this work.
		* Minor
			- N/A

+ Significance
	- Strengths
		- The reported results would seem to confirm that, up to suitable hyperparameter tuning, the proposed curiosity signals do enhance exploration for RL fine-tuning to result in improved performance beyond GRPO/PPO.
	- Weaknesses
		* Major
			- Across the tasks, the gains in performance seem quite small numerically. Moreover, there is no reporting of how many trials/random seeds are used to obtain these results leaving ambiguous the questions of whether the proposed curiosity metrics consistently enchance exploration in a way that yields statistically-significant performance improvements. Also, given the small size of improvements, whether the proposed curiosity signals are worthwhile given the potentially laborious grid search needed to find suitable hyperparameter values.
		* Minor
			- N/A


+ Final Remarks
	- Overall, I have identified critical issues with this paper on the axes of quality, significance, and originality. While the initial idea has some potential, there is much more work needed on the technical details and presentation as well as the empirical support for the method before publication.

**Weaknesses:**

Please see above.

**Questions:**

Please see above.

---

> ### Author Response · Authors · 2025-11-21
> **Response to Quality Major Weaknesses Part I**
>
> We sincerely appreciate the reviewer’s recognition of our insights, the clarity of the manuscript, and the novelty of CDE. These encouraging comments are invaluable to us. We address each concern in detail below.
>
> **MQ1: The necessity of discussions of curiosity**
>
> First, we would like to clarify that our reference to childhood development is used purely as a metaphor to convey the intuition behind the LLM RL training process. As stated in the paper, “LLM constitutes a familiar versus a novel reasoning pattern … paralleling early childhood development,” which is intended solely as an analogy rather than a scientific claim and we do not believe scientific evidence is required.
>
> Such human-inspired analogies are common and widely accepted in the LLM reasoning literature. Terms like reasoning, chain-of-thought, or planning are themselves metaphorical borrowings from psychology and cognitive science. Similarly, classic curiosity-driven RL works also rely on intuitive metaphors to motivate intrinsic motivation. For example, the ICM paper [1] introduces curiosity by comparing an RL agent to “a three-year-old entertaining herself in a playground,” which also serves to convey intuition rather than to assert a scientific fact.
>
> In our paper, this analogy plays an important role in motivating the notion of actor-wise and critic-wise curiosity bonuses, helping readers—especially those from NLP backgrounds without RL experience—build an intuitive understanding of these signals. Simply citing prior curiosity-driven RL work without providing such intuition would be less accessible.
>
> **MQ2: Why shape advantage in PPO**
>
> In our paper, we choose to shape the advantage in PPO primarily due to concerns about algorithmic stability. Compared with traditional RL environments, LLM reasoning involves extremely long horizons, sparse reward, and an enormous state–action space. Directly modifying the reward not only perturbs the optimization objective but also injects noise into value estimation. In contrast, shaping the advantage introduces only a modest modification: it preserves the value function while encouraging the actor to explore under-explored regions. Besides, the clipping mechanism ensures that the exploration signal remains controlled. By comparison, injecting bonuses into intermediate steps would inevitably dominate the reward (zero for intermediate steps) and may lead to instability.
>
> We also note that advantage shaping is widely adopted in recent RLVR work. For example, [2], an RND-style intrinsic motivation method discussed in Section 5, follows the same design choice and explicitly expresses concerns about traditional reward-shaping approaches in their Section 2.2.3. Likewise, [3,4] employ similar strategies based on shaping or truncating the advantage.
>
>
> **MQ3: High PPL responses may correspond to low quality**
>
> We acknowledge that high PPL can arise from either genuine under-exploration or from previously suppressed, sub-optimal trajectories. However, this ambiguity is inherent to most intrinsic exploration signals (e.g., count-based or curiosity-driven methods), since low visitation or high PPL can both result from policy updates decreasing the probability of certain actions. In all such frameworks, trial and error serves as the ultimate filter: even if a sub-optimal response initially triggers high PPL, after severl attemps the RL optimization will eventually suppress it due to low extrinsic value.
>
> Crucially, our design in Eq. (2) further safeguards against this ambiguity. Importantly, we never rely on PPL alone. The curiosity bonus is always combined with the extrinsic label reward, and the truncation rule in Eq. (2) ensures that the final shaped reward always preserves the sign of the label reward. Consequently, a high-PPL but incorrect response can never outweigh a correct one.

---

> ### Author Response · Authors · 2025-11-21
> **Response to Quality Major Weaknesses Part II**
>
> **MQ4: Proof of Theorem 3.2**
>
> We appreciate the reviewer’s concern regarding the level of detail in the proof of Theorem 3.2. Some intermediate steps were initially omitted because they follow standard derivations in RL theory, but we agree that this may cause confusion. Importantly, these omissions do not affect the correctness of the argument. In the revised PDF, we have expanded the proof with full intermediate steps and explicit derivations.
>
> We also respectfully disagree with the reviewer’s characterization that the proof is "an amalgamation of equations" copied from other RL theory papers. Our analysis indeed adopts the standard linear-MDP framework used in works such as LSVI-UCB, but the convergence analysis of the standard deviation across bootstrap value heads is specific to our method. To the best of our knowledge, the most related work is [5]. However, their analysis is conducted under a Bayesian framework, whereas ours adopts a frequentist perspective, and they do not explicitly examine the convergence behavior of the bootstrap approach.
>
> Finally, our theoretical result is not intended to add "mathiness" to justify a heuristic. The result formalizes the behavior of a bootstrap-based exploration mechanism, which is a well-established and widely used statistical technique. While we acknowledge that the initial draft presented the proof with overly compressed steps, the theorem itself is substantive and rooted in standard statistical principles. We hope that the expanded and clarified version will address the reviewer’s concerns.
>
> Below are point to point response to the points pointed by the reviewer.
>
> 1. The order $ \mid \Lambda_{n,t}^{-1} \mid_{op} = O_p(\frac{1}{n})$ is a direct consequence of the definition of $\Lambda_{n,t}^{-1}$ and the definition of operator norm.
>
> 2. Expansion in L895 is a consequence of the Woodbury matrix identity. In fact, it is the "recursive structure" listed in the "Inverse of a sum" section on wikipedia [6]. We have derived a step-to-step derivation of such expansion in the updated PDF.
>
> 3. Equation 4 follows directly from expanding the definitions and rearranging terms. In particular, substituting the definitions of $\Delta b$, $b$, $b_s$, and $G_{i,t}$ into $\Lambda_{n,t}^{-1}\Delta b$ produces both the leading term in Equation 4 and the term $\phi^\top \Lambda_{n,t}^{-1}\Delta\Sigma, w^*_t$. We believe the current level of detail is appropriate and do not plan to add further derivations.
>
> 4. The notation $ \mathrm{Var}^* $ was intended to denote the conditional variance given $\\{\phi_{i,t}, \epsilon_{i,t}\\}_{i=1}^n$. To avoid confusion, we have removed this notation in the revised PDF accordingly.

---

> ### Author Response · Authors · 2025-11-21
> **Response to Quality Minor Weaknesses Part I**
>
> **mQ1: The use of word "perplexity".**
>
> We use the term “perplexity” because it is the standard and widely recognized terminology in the NLP community, whereas
> "surprisal" is less commonly used in practice. As noted in our footnote, our quantity corresponds to the logarithm of perplexity, and several reviewers explicitly acknowledged this naming as understandable. To avoid further ambiguity, we will move the footnote to the first page as suggested by reviewer fSmQ.
>
> **mQ2: Necessity of Equation 2**
>
> We thank the reviewer for noting that the explanation and naming of the hyperparameters is helpful. While it is mathematically possible to consolidate the terms involving $\omega/\kappa$ and $\alpha\kappa$, doing so would introduce undesirable coupling between parameters that play different roles. In our design, $\omega$ decays over the course of training while the other parameters remain fixed. If these terms were merged, both resulting coefficients needs to vary (e.g., $\omega/\kappa$ increasing while $\alpha\kappa$ decreases), making the equation harder to interpret, and complicating implementation in practice.
>
> Regarding the suggestion to rewrite Equation 2 as $r + \lambda B$, this is not compatible because the bonus is applied through a non-linear truncation, $\min(\cdot,\cdot)$, which fundamentally prevents it from being expressed as a linear additive term.
>
>
> **mQ3: Increasing probability mass for incorrect responses**
>
> Yes, that is the key reason we introduce the parameter $\kappa$ and the truncation mechanism in Equation 2. As the explanations in our response to Q3, it ensures a high log-PPL but incorrect response can never outweigh a correct one.
>
> **mQ4: Theorem 3.1 shouldn't be called theorem.**
>
> We thank the reviewer for the helpful suggestion. We will rename Theorem 3.1 to Proposition 3.1 in the revised version.
>
> **mQ5: Clarification of Posterior Distribution**
>
> We clarify that the "posterior distribution" refers to the distribution over the value function $V_\theta$, conditioned on the collected data.
>
> From a frequentist's view, the value estimator is a random variable subject to the stochasticity of the data sampling and optimization. The empirical distribution of our value heads approximates the sampling distribution of the value estimator. This distribution is essentially linked to the posterior distribution in the Bayesian perspective. We will intuitively justify such link though the following Bernstein-von Mises (BvM) theorem [7].
>
> Let $\\{P_\theta : \theta \in \Theta\\}$ be a parametric model and let data $X_{1:n}$ be i.i.d. samples from $P_{\theta^\star}$. Under standard regularity conditions of the prior, the BvM theorem states that the posterior is asymptotically Gaussian and centered near the estimator $\hat{\theta}_n$:
>
> $$
>   p(\theta \mid X_{1:n})
>   \approx
>   \mathcal N\bigl(\hat\theta_n, I(\theta^\star)^{-1}/n\bigr)
>   \quad\text{for large } n.
> $$
> Simultaneously, frequentist theory yields the asymptotic normality of the MLE $\hat{\theta}_n$ around the true parameter $\theta^\star$:
>
> $$
>   \hat\theta_n
>   \approx
>   \mathcal N\bigl(\theta^\star, I(\theta^\star)^{-1}/n\bigr)
>   \quad\text{for large } n.
> $$
> Since the difference between centering at $\theta^\star$ and $\hat\theta_n$ is asymptotically negligible ($O_p(n^{-1/2})$), the posterior and the sampling distribution are asymptotically equivalent to the first order. By the Delta method, this equivalence extends to the value function. For a given state $s$, the distribution of the estimator $V_{\hat{\theta}_n}(s)$ is:
>
> $$
> V_{\hat \theta_n}(s) \approx \mathcal{N} (  V_{\theta^\star}(s),  \frac{1}{n}  \nabla_\theta V_{\theta}(s) \mid_{\theta=\theta^\star}^{\top}  I(\theta^\star)^{-1}  \nabla_\theta V_\theta(s)\mid_{\theta=\theta^\star}).
> $$
> This distribution is asymptotically equivalent to the Bayesian posterior distribution of the value function.
>
> Our multi-head critic leverages this equivalence. Each head is trained on a different bootstrap resample of the data. Consequently, the ensemble predictions $\\{V_{\hat{\theta}^k}(s)\\}_{k=1}^K$ approximate samples from the posterior predictive distribution of the value. This allows us to perform uncertainty-based exploration without explicit Bayesian inference.
>
> Finally, we want to address that such view has been used extensively in previous bootstrap RL methods [8,9] to reduce the complexity of exact Bayesian RL approach.

---

> ### Author Response · Authors · 2025-11-21
> **Response to Quality Minor Weaknesses Part II**
>
> **mQ6: Comparison to Max-Entropy and Soft Actor Critic.**
>
> We thank the reviewer for acknowledging that our log-PPL bonus meaningfully extends traditional entropy regularization. However, comparing against max-entropy RL algorithms such as discrete SAC [10] is unfortunately not feasible in the LLM setting.
>
> In particular, discrete SAC requires a Q-network that outputs a value for every action in the action space. The actor update then computes gradients that enumerate over all actions. While this is practical for small discrete action spaces, it becomes computationally intractable for LLMs, where $|A|\approx 50{,}000$. For this reason, SAC-style max-ent RL has not been applied to LLM training in any prior work.
>
> A more appropriate and tractable baseline is adding an entropy regularization term to GRPO/PPO. We have now included this baseline in the revision, and the corresponding results are reported below:
>
> | **Model**                 | **MATH** Avg@1 | **AMC23** Avg@16 | **AMC23** Pass@16 | **AIME24** Avg@16 | **AIME24** Pass@16 | **AIME25** Avg@16 | **AIME25** Pass@16 | **Avg** |
> | ------------------------- | -------------- | ---------------- | ----------------- | ----------------- | ------------------ | ----------------- | ------------------ | ------- |
> | Qwen3-4B-Base-GRPO        | 87.3           | 63.6             | 89.1              | 20.8              | 41.9               | 21.0              | 39.2               | 48.2    |
> | → **w/ Entropy bonus**    | 86.8           | 64.3             | 89.7              | 21.8              | 39.4               | 21.2              | 41.1               | 48.5    |
> | → **w/ PPL bonus (Ours)** | 87.7           | 67.8             | 89.5              | 23.3              | 48.5               | 23.5              | 42.5               | 50.6    |
> | Qwen3-4B-Base-PPO         | 86.6           | 64.1             | 87.2              | 17.8              | 36.0               | 17.5              | 33.7               | 46.5    |
> | → **w/ Entropy bonus**    | 83.9           | 66.3             | 87.5              | 17.9              | 30.6               | 19.5              | 33.0               | 46.9    |
> | → **w/ 4 Heads (Ours)**   | 87.3           | 63.9             | 87.9              | 21.5              | 35.5               | 21.5              | 45.5               | 48.5    |
>
> **mQ7: Clarification of value ensemble.**
>
> As we have clarified in response to Q8. The empirical distribution formed by the value heads approximates the sample distribution of the value estimator. It is natural to use the empirical mean to estimate the value estimator, which provide a more robust value estimate than relying on a single head. Such approach is also used in previous works, where Q-Ensemble [8] use empirical mean of Q heads plus empirical std to perform UCB-like action selection. Also in recent RLVR work[4], they also use the ensemble of multiple head to serve as value estimation.

---

> ### Author Response · Authors · 2025-11-21
> **Response to Clarity**
>
> **C1: The necessity of Hashing based technique**
>
> We agree that the hashing-based technique is not the core contribution of our paper, but we have already constrained the discussion to a brief summary while leaving the full setup to the appendix.
>
> However, we believe retaining this discussion is important for contextualizing why CDE is effective. Presenting the shortcomings of hashing clarifies why naive count-based methods perform poorly in the LLM setting. This contrast strengthens the contribution of CDE by illustrating the specific failure modes that our method overcomes.
>
> Moreover, this comparison has been found helpful by other reviewers (e.g., reviewer fSmQ noted that the SimHash analysis is informative). It helps readers outside RL community to better understand different exploration approaches. Therefore, we believe keeping this discussion adds value without detracting from the clarity of the main paper.

---

> ### Author Response · Authors · 2025-11-21
> **Response to Originality**
>
> **O1: Literature and baseline of Intrinsic Reward RLVR.**
>
> First, we would like to clarify that we do not claim to be the first to introduce intrinsic-motivation–based exploration in the RLVR setting. We have conducted a thorough literature review of intrinsic-reward methods for RLVR and have discussed all relevant works we are aware of. If we have inadvertently missed any references, we would be happy to include them in the revised manuscript.
>
> Regarding baseline comparisons, to the best of our knowledge, only one curiosity-driven RLVR method currently provides publicly available code: i-MENTOR [2]. This work is a very recent arXiv preprint whose code was released in July, and thus—per conference policy—was not required as a baseline.
>
> That said, we agree that including this baseline strengthens our submission. We have implemented and evaluated i-MENTOR in our framework. As shown below, i-MENTOR-GRPO produces a modest improvement over GRPO, while i-MENTOR-PPO does not improve upon PPO. In contrast, our method outperforms i-MENTOR across both settings, further demonstrating the effectiveness and robustness of our approach.
>
> | **Model**                 | MATH Avg@1 | AMC23 Avg@16 | AMC23 Pass@16 | AIME24 Avg@16 | AIME24 Pass@16 | AIME25 Avg@16 | AIME25 Pass@16 | **Avg** |
> | ------------------------- | ---------- | ------------ | ------------- | ------------- | -------------- | ------------- | -------------- | ------- |
> | Qwen3-4B-Base-GRPO        | 87.3       | 63.6         | 89.1          | 20.8          | 41.9           | 21.0          | 39.2           | 48.2    |
> | → **i-MENTOR-GRPO**       | 87.6       | 63.2         | 89.1          | 22.5          | 39.3           | 23.0          | 40.4           | 49.1    |
> | → **w/ PPL bonus (Ours)** | 87.7       | 67.8         | 89.5          | 23.3          | 48.5           | 23.5          | 42.5           | 50.6    |
> | Qwen3-4B-Base-PPO         | 86.6       | 64.1         | 87.2          | 17.8          | 36.0           | 17.5          | 33.7           | 46.5    |
> | → **i-MENTOR-PPO**        | 85.8       | 60.9         | 85.9          | 18.9          | 39.4           | 19.6          | 36.5           | 46.3    |
> | → **w/ 4 Heads (Ours)**   | 87.3       | 63.9         | 87.9          | 21.5          | 35.5           | 21.5          | 45.5           | 48.5    |
>
> **O2: More baseline beyond intrinsic reward.**
>
> As discussed in our response to mQ6, we have added GRPO/PPO with entropy bonus as an additional baseline to cover exploration mechanisms outside intrinsic-motivation–based methods. Furthermore, as noted in our response to O1, we have also incorporated i-MENTOR into our empirical evaluation. Together, these additions broaden the set of baselines and provides a more comprehensive empirical comparison.
>
> As for the two references provided by the reviewer, neither is appropriate because both address fundamentally different problems. "Efficient Exploration for LLMs" focuses on inference-time response selection—i.e., deciding which generated samples should be sent to human annotators during RLHF data collection. It does not aim to improve exploration during RL training. "Hindsight Merging", on the other hand, introduces a "post-training model-merging" procedure that fuses a trained model with its base model. Given these substantive differences in both purpose and setting, neither work constitutes a suitable baseline for our RLVR training problem.

---

> ### Author Response · Authors · 2025-11-21
> **Response to Significance**
>
> **S1: Request to number of trails/random seeds and effectiveness of CDE**
>
> We appreciate the reviewer’s suggestion regarding multi-seed evaluation. However, RLVR training on LLMs is extremely expensive: a single run typically requires 8×A100 GPUs for 2–3 days, making multi-seed sweeps impractical. Consequently, prior state-of-the-art RLVR works (e.g., DeepSeek-R1 [11], DAPO [12], PRIME [13]) also report single-seed results without confidence intervals, and we follow the same standard practice. In addition, our Avg@16 evaluation in Table 2 and the accuracy curve in Figure 6 both show consistent improvement throughout training, accounting for stochasticity in training and evaluation and providing strong evidence for the robustness of our method.

---

> ### Author Response · Authors · 2025-11-21
> **Reference**
>
> [1] Pathak, Deepak, et al. "Curiosity-driven exploration by self-supervised prediction." International conference on machine learning. PMLR, 2017.
>
> [2] Gao, Jingtong, et al. "Navigate the unknown: Enhancing llm reasoning with intrinsic motivation guided exploration." arXiv preprint arXiv:2505.17621 (2025).
>
> [3] Cheng, Daixuan, et al. "Reasoning with exploration: An entropy perspective." arXiv preprint arXiv:2506.14758 (2025).
>
> [4] Liu, Jiashun, et al. "Asymmetric Proximal Policy Optimization: mini-critics boost LLM reasoning." arXiv preprint arXiv:2510.01656 (2025).
>
> [5] Bai, Chenjia, et al. "Principled exploration via optimistic bootstrapping and backward induction." International Conference on Machine Learning. PMLR, 2021.
>
> [6] https://en.wikipedia.org/wiki/Woodbury_matrix_identity
>
> [7] https://en.wikipedia.org/wiki/Bernstein-von_Mises_theorem
>
> [8] Chen, Richard Y., et al. "Ucb exploration via q-ensembles." arXiv preprint arXiv:1706.01502 (2017).
>
> [9] Osband, Ian, et al. "Deep exploration via bootstrapped DQN." Advances in neural information processing systems 29 (2016).
>
> [10] Christodoulou, Petros. "Soft actor-critic for discrete action settings." arXiv preprint arXiv:1910.07207 (2019).
>
> [11] Guo, Daya, et al. "Deepseek-r1 incentivizes reasoning in llms through reinforcement learning." Nature 645.8081 (2025): 633-638.
>
> [12] Yu, Qiying, et al. "Dapo: An open-source llm reinforcement learning system at scale." arXiv preprint arXiv:2503.14476 (2025).
>
> [13] Cui, Ganqu, et al. "Process reinforcement through implicit rewards." arXiv preprint arXiv:2502.01456 (2025).

---

> > ### Comment · Reviewer_XcFv · 2025-11-28
> >
> > I thank the authors for their time and effort in crafting a rebuttal response to my review.
> >
> > > our reference to childhood development is used purely as a metaphor to convey the intuition behind the LLM RL training process.
> >
> >
> >
> >
> > Regardless of whether or not it was intended, the phrase "Our approach is inspired by early cognitive development (Chu & Schulz, 2020)" (L157-158) hints at the possibility of some deep, substantive connection between the authors' approach and developmental psychology --- there isn't one. If the authors' use of curiosity was purely metaphorical and done in such a way that it was abundantly clear no suggestion of being actually related to cognitive science is made, I would have no issue.
> >
> > > Such human-inspired analogies are common and widely accepted in the LLM reasoning literature.
> >
> >
> >
> >
> > This kind of hyperbole is largely just as pointless in these papers as it is in the authors' work and the desire to anthropomorphize LLMs has no bearing on the scientific contributions offered in this work.
> >
> > > this analogy plays an important role in motivating the notion of actor-wise and critic-wise curiosity bonuses, helping readers—especially those from NLP backgrounds without RL experience
> >
> >
> >
> >
> > This concern about accessibility doesn't really make sense. Why would the authors consider readers "from NLP backgrounds without RL experience" a part of the target audience for this work? (This is a rhetorical question that the authors should not feel compelled to provide an actual answer to.)
> >
> > > Compared with traditional RL environments, LLM reasoning involves extremely long horizons, sparse reward, and an enormous state–action space.
> >
> >
> >
> >
> > Just as a matter of fact, while the authors point about "extremely long horizons" is well taken, it should be noted that there is no shortage of "traditional RL environments" with sparse rewards and enormous state-action spaces (in the case of continuous-control problems, state-action spaces infinitely larger than those considered in LLM reasoning).
> >
> > > Directly modifying the reward not only perturbs the optimization objective but also injects noise into value estimation.
> >
> >
> >
> >
> > It's not at all clear what this means. "Perturbs the optimization objective"...how? The standard approach is to only modify the reward function and leave everything else unchanged, so the form of the objective is the same but feedback signal optimized obviously differs. Also, what noise is being injected into value function estimation?
> >
> > > injecting bonuses into intermediate steps would inevitably dominate the reward (zero for intermediate steps) and may lead to instability.
> >
> >
> >
> >
> > I agree that the difference in scales between the original (extrinsic) reward and the newly added intrinsic reward could lead to an issue where the latter dominates the former. This is precisely why it is quite common to introduce a hyperparameter in the form of a weighting coefficient to ensure these rewards operate on similar scales and one cannot be ignored in favor of another; it is presumably this exact intuition that prompts the authors to introduce their $\omega$ and $\alpha$ hyperparameters in their modified advantage function.
> >
> > > advantage shaping is widely adopted in recent RLVR work.
> >
> >
> >
> >
> > While I understand that authors may have just been following an emerging trend/hack, this doesn't change the original point of my critique that this choice to redefine the advantage function deviates from how intrinsic rewards are commonly utilized in the RL literature. I don't think it is unreasonable to expect that the authors have some kind of justification for why. The attempt to articulate a mathematical one above due to perturbations and injections isn't effective, so perhaps the simpler thing to do would be to run an experiment confirming that the standard way of incorporating intrinsic motivation isn't fruitful empirically.
> >
> > > this ambiguity is inherent to most intrinsic exploration signals (e.g., count-based or curiosity-driven methods)
> >
> >
> >
> >
> > The more accurate phrasing of this response from the authors is that the issue I've raised is a deficiency of curiosity-driven approaches to exploration that perhaps the authors have inherited. Count-based methods are a subset of exploration methods in RL driven by the principle of "optimism in the face of uncertainty." The initial optimism injected to the value estimates of all state-action pairs at the very start of learning is precisely what avoids this error as, while the visitation is low, the optimistic inflation of value persists and, once visitation is low, there is some kind of high-probability guarantee that the state-action pair has been explored sufficiently.

---

> > > ### Comment · Reviewer_XcFv · 2025-11-28
> > >
> > > > In all such frameworks, trial and error serves as the ultimate filter: even if a sub-optimal response initially triggers high PPL, after severl attemps the RL optimization will eventually suppress it due to low extrinsic value.
> > >
> > >
> > >
> > >
> > > This is a catchy opening to the sentence, but the following reasoning doesn't quite land for me. The issue is correct --- namely, that a high surprisal (still not perplexity) may indicate either an under-explored response or a sub-optimal response. The authors' response is that their underlying choice of policy optimization algorithm will essentially figure it out from low value estimates generated by the critic. How exactly would that occur? I agree it's nice to hope that it will happen, but mathematically what about the authors' proposed setup will make it so? A critic is trained to estimate the value function induced by the current policy and the authors' surprisal bonus modifies the reward function, which in turn changes what the critic aims to estimate (high surprisal included).
> > >
> > > > our design in Eq. (2) further safeguards against this ambiguity.
> > >
> > >
> > > > a high-PPL but incorrect response can never outweigh a correct one.
> > >
> > >
> > >
> > >
> > > I assume by "our design" the authors refer to three hyperparameters colored in Equation 2. While I agree with their statement, this doesn't really seem like the makings of a well-designed intrinsic reward. It only sometimes gives the exploratory indicator that we actually want and, for those other incorrect times, an agent designer needs to play with three different hyperparameters to deal with it.
> > >
> > > > Proof of Theorem 3.2
> > >
> > >
> > >
> > >
> > > I'm glad the authors took the time to finish their proof of Theorem 3.2. It's rather shocking that roughly a page of mathematical steps and justifications were left out in the original submission. Everything appears to be in order now.
> > >
> > > > We use the term “perplexity” because it is the standard and widely recognized terminology in the NLP community, whereas "surprisal" is less commonly used in practice.
> > >
> > >
> > >
> > >
> > > While I understand the marketing implications of this to the NLP community, distinct words have distinct meanings and so we might as well use them. It would be more sensible for the authors to acknowledge their bonus as surprisal and then give a footnote to those readers who might only be familiar with perpexlity.
> > >
> > > > While it is mathematically possible to consolidate the terms involving and , doing so would introduce undesirable coupling between parameters that play different roles.
> > >
> > >
> > >
> > >
> > > If these things truly can't be consolidated, then it would be logical to have some kind of sensitivity analyses across the various hyperparameters to give readers an indication of how laborious the hyperparameter tuning process might be on account of using the proposed method.
> > >
> > > > A more appropriate and tractable baseline is adding an entropy regularization term to GRPO/PPO.
> > >
> > >
> > > > i-MENTOR-GRPO produces a modest improvement over GRPO, while i-MENTOR-PPO does not improve upon PPO. In contrast, our method outperforms i-MENTOR across both settings
> > >
> > >
> > >
> > > While I appreciate the authors' addition of the entropy regularization and i-MENTOR baselines, there is still something left to be desired from the empirical evaluation. As I look across these tables of numbers from single-seed runs with no way to assess whether these findings are consistent or replicable, I can't tell if this empirical paper has made a significant advance or not. Many of the improvements between the proposed method and baseline methods are quite small. Rather than trying to read the tea leaves here, I'll simply ask the authors. Looking at these quantitative results, how have the authors concluded that they have achieved a meaningful advance in RLVR through their proposed curiosity-driven exploration approach?

---

> ### Comment · Reviewer_XcFv · 2025-11-28
>
> > neither work constitutes a suitable baseline for our RLVR training problem.
>
>
> > i-MENTOR [2]. This work is a very recent arXiv preprint whose code was released in July, and thus—per conference policy—was not required as a baseline.
>
>
> > We are on the same page that comparisons to contemporaneous work are not mandatory, especially for papers [1] and [2] which appeared on arXiv in late August 2026, concurrent with our submission period.
>
>
>
>
> Agreed, these were pointers to existing works that have already considered the problem of exploration for improving response diversity in LLMs through other means. Still, for the particular RLVR problem considered, surely there are many approaches that have tried to address exploration along other avenues (not necessarily dependent on curiosity) which precede the ICLR submission window? I'm not going to do the authors' literature review for them, so let me ask in a different way. Are the authors claiming that there are no prior methods (which cannot be construed as contemporaneous) which tackle the RLVR exploration challenge studied in this work? If the answer is genuinely yes, then this work is truly groundbreaking in being the first to recognize and attempt to address the exploration challenge as it pertains to RL fine-tuning of LLMs. If the answer is no, then presumably there are two classes of alternative baseline methods: (1) those that rely on curiosity or intrinsic motivation, but formulate the intrinsic reward in a way that differs from the authors' submission or (2) those that attempt to tackle exploration in a fundamentally different, orthogonal manner to intrinsic motivation. It sounds like DACE and i-MENTOR belong to the former category, so the i-MENTOR results provide one (let's assume, strong) baseline comparison. What about a strong baseline comparison from the latter category? Are there truly no candidates?

---

> > ### Author Response · Authors · 2025-11-28
> >
> > Thanks to the reviewer for your thoughtful reply. We are glad that among the 15 concerns you raised, many have now been resolved through our rebuttal, with only a few requiring further clarification. We are actively working on addressing those remaining points. Given that a substantial portion of your concerns have been addressed, we would be grateful if you could consider updating your evaluation accordingly.
> >
> > Wish you a happy Thanksgiving!

---

> > ### Author Response · Authors · 2025-12-01
> > **Final response part I**
> >
> > Thanks the reviewer for your reply. We are glad that among the 15 concerns you raised, many (MQ4,mQ2,3,4,5,6,7,O1) have now been resolved through our rebuttal.
> >
> > For clarity, I will summarize and list the remaining concerns of the reviewer and address them one-by-one.
> >
> > ## **1. The human cognitive metaphor and the use of word Perplexity**
> >
> > We introduced the metaphor purely as an intuitive aid for readers who may not have a strong RL background, especially those unfamiliar with intrinsic-motivation–based exploration. We believe this clarification is necessary given that many LLM reasoning researchers come from the NLP community rather than traditional RL.
> >
> > We use Perplexity instead of surprisal because PPL is a widely understood and accepted metric in the NLP community, including by other reviewers.
> >
> > We respectfully disagree that this wording warrants strong criticism, and we believe such **stylistic preferences should not lead to strong rejection of our submission**.
> >
> > ## **2. Why shaping the Advantage instead of shaping reward in traditional RL**
> >
> > Compared with traditional reward shaping in RL, shaping the advantage offers two key benefits:
> >
> > 1. Better control of the bonus via clipping.
> >    Injecting bonuses into intermediate rewards would dominate the signal in RLVR, since intermediate rewards are 0. Advantage shaping avoids this issue and keeps the bonus under control.
> > 2. Cleaner learning targets for the critic.
> >    Modifying intermediate rewards would shift the critic’s target to final reward + accumulated bonuses, complicating value estimation. By shaping the advantage instead, the critic continues to learn solely from the labeled reward, preserving its stability.
> >
> > **We have conducted addtional experiments using reward shaping as requested by the reviewer**. We tested shaping the reward in PPO using the bootstrap bonus under multiple bonus weights (1, 0.1, 0.01), but none of these configurations yielded stable training. **In all cases, the critic failed to converge, the actor’s response length quickly saturated at the maximum limit, and performance collapsed to zero.**
> >
> > Overall, these empirical observations, the logical considerations above, and the widespread use of this modification in recent RLVR works [1–3] provide a comprehensive justification for our choice of advantage shaping over traditional reward shaping.
> >
> > ## **3. Explaination to incorrect High PPL responses will be suppressed**
> >
> > Our bonus-truncation mechanism ensures that incorrect high-PPL responses always receive a **negative reward**, strictly lower than that of correct responses. Compared with correct outputs, low-quality (incorrect) high-PPL responses are explicitly pushed downward.
> >
> > We would also like to clarify a misunderstanding: **nowhere** in the manuscript or rebuttal did we attribute this behavior to “low value estimates generated by the critic.” The effect arises solely from the **verifiable reward of –1** assigned to incorrect answers.

---

> > ### Author Response · Authors · 2025-12-01
> > **Final response part II**
> >
> > ## **4. Request for sensitivity analysis**
> >
> > We have also included sensitivity analyses on the remaining hyperparameters. Together with the original analyses on the annealing schedule and sampling fraction, these results offer a comprehensive characterization of CDE’s robustness.
> >
> > | **Model**                                 | **MATH** Avg@1 | **AMC23** Avg@16 | **AMC23** Pass@16 | **AIME24** Avg@16 | **AIME24** Pass@16 | **AIME25** Avg@16 | **AIME25** Pass@16 | **Avg** |
> > | ----------------------------------------- | -------------- | ---------------- | ----------------- | ----------------- | ------------------ | ----------------- | ------------------ | ------- |
> > | Qwen3-4B-Base-PPO                         | 86.6           | 64.1             | 87.2              | 17.8              | 36.0               | 17.5              | 33.7               | 46.5    |
> > | → **$\kappa=3 $,$\alpha=0.5$** (Original) | 87.3           | 63.9             | 87.9              | 21.5              | 35.5               | 21.5              | 45.5               | 48.5    |
> > | → **$\kappa=1$,$\alpha=1$**               | 86.5           | 65.0             | 89.5              | 20.9              | 42.5               | 20.7              | 39.8               | 48.3    |
> >
> > | **Model**                               | **MATH** Avg@1 | **AMC23** Avg@16 | **AMC23** Pass@16 | **AIME24** Avg@16 | **AIME24** Pass@16 | **AIME25** Avg@16 | **AIME25** Pass@16 | **Avg** |
> > | --------------------------------------- | -------------- | ---------------- | ----------------- | ----------------- | ------------------ | ----------------- | ------------------ | ------- |
> > | Qwen3-4B-Base-GRPO                      | 87.3           | 63.6             | 89.1              | 20.8              | 41.9               | 21.0              | 39.2               | 48.2    |
> > | → **$\kappa=3 $,$\alpha=1$** (Original) | 87.7           | 67.8             | 89.5              | 23.3              | 48.5               | 23.5              | 42.5               | 50.6    |
> > | → **$\kappa=1$,$\alpha=1$**             | 20.9           | 11.1             | 54.6              | 1.5               | 12.7               | 1.3               | 11.4               | 8.67    |
> > | → **$\kappa=2$,$\alpha=1$**             | 87.6           | 66.5             | 86.9              | 22.3              | 42.5               | 21.6              | 40.1               | 49.5    |
> > | → **$\kappa=4$,$\alpha=1$**             | 88.6           | 68.2             | 90.1              | 23.6              | 46.6               | 22.7              | 40.8               | 50.8    |
> >
> > The critic-wise bonus is generally robust to hyperparameter choices; setting both $\kappa$ and $\alpha$ to 1 produces nearly identical performance. The actor-wise bonus is relatively more sensitive. Generally, to get realative good performance, the clipping fraction $\kappa$ should not be set to small, i.e. the bonus should be constrained to be not exceeding at least 50/% of the labeled reward.
> >
> > ## **5. Request addtional baselines and question the significance of CDE**
> >
> > First, we would like to clarify that we never claimed to be the first to introduce exploration in RLVR. We have explicitly cited relevant prior works and **have already added two addtional baselines (i-MENTOR and entropy bonuses) as you suggested during rebuttal**, covering both two sides as you pointed out.
> >
> > However, **none of RLVR exploration works even i-MENTOR were published in peer-reviewed venues before July 24, 2025**. Per **ICLR policy**, authors are not required to compare against unpublished arXiv-only papers. Therefore, we respectfully **disagree that requesting such baselines constitutes a valid reason for a strong rejection**.
> >
> > Regarding significance of our results, we conducted extensive experiments across various model family, benchmarks, which took nearly two weeks to complete. The results consistently show that our method outperforms both intrinsic-motivation and entropy-based baselines. The significance of CDE is recognized by all reviewers and we believe these comprehensive results provide sufficient evidence of the contribution and significance of our approach.
> >
> > ## Reference
> > [1] Gao, Jingtong, et al. "Navigate the unknown: Enhancing llm reasoning with intrinsic motivation guided exploration." arXiv preprint arXiv:2505.17621 (2025).
> >
> > [2] Cheng, Daixuan, et al. "Reasoning with exploration: An entropy perspective." arXiv preprint arXiv:2506.14758 (2025).
> >
> > [3] Liu, Jiashun, et al. "Asymmetric Proximal Policy Optimization: mini-critics boost LLM reasoning." arXiv preprint arXiv:2510.01656 (2025).

---

### Author Response · Authors · 2025-12-01
**Summary**

Dear All Reviewers and AC,

We would like to express our sincere gratitude to all reviewers for their insightful suggestions and constructive comments, which have been immensely helpful to us. Following the reviewers’ feedback, we have incorporated additional experiments and made substantial revisions to strengthen the manuscript. For your convenience, we summarize the main updates below:

- **More detailed proof of Theorem 3.2** *XcFv*
- **Additional intrinsic motivation baseline(i-Mentor)** *XcFv,fSmQ,KJQc*
- **Additional entropy bonus baseline** *XcFv*
- **New model family(llama 3.2 3B Instruct)** *fSmQ*
- **More detailed Ablation study** *XcFv,fSmQ*
- **Combining multi-head with PPL bonus** *fSmQ,KJQc,XET9*
- **Computation budget of Multi-head critic** *KJQc*

We would like to once again thank all reviewers for their time and thoughtful evaluations. Your insightful comments, both positive and critical, have significantly contributed to strengthening our manuscript.

Best regards,

Authors of CDE

---

### Meta-Review · Area_Chair_xKvB · 2026-01-07

**Summary:**

This paper proposes a "curiosity-driven" approach for exploration in RLVR, reshaping the advantage estimate to give a a bonus for the log-perplexity/surprisal in the actor or the variance of value estimates from a multi-head critic.

Most reviewers were mildly positive on the paper, with the exception of XcFv who was strongly negative. In post-review discussion that has been preserved for me as the new AC, this reviewer stated that they maintained their overall opinion, with an updated score of "reject" rather than "strong reject"; I do not have access to their specific responses to the final review, however, meaning that in this unusual ICLR review process I must judge based on my reading of the back-and-forth and the paper itself. To my mind, many of XcFv's complaints have been resolved by the rebuttal, while others of those that remain are not particularly convincing to me (e.g. the level of terminological complaint about surprisal vs perplexity; I would perhaps prefer if you consistently referred to it as the "log-perplexity", which addresses both sides, but I do not think this is an important distinction at all in this context).

One meaningful remaining concern from XcFv and other reviewers, I think, is the limited comparison to other approaches for curiosity in general RL settings applied to this case. While I expect that there exist some older papers than the recent arXiv papers you've identified that could be fairly directly applied to this setting, I am not knowledgeable enough in this area to know offhand what they are, and given that the reviewers have not identified such approaches either, I do not think this is sufficient to warrant rejection.

I am also sensitive to both the concern about single runs and the reality that these training runs are quite expensive, and thus running multiple seeds for all settings is computationally prohibitive. It is widely recognized in "standard" RL settings that single-seed experiments can be quite misleading (see e.g. https://arxiv.org/abs/1709.06560), and yet, despite the widely known drawbacks, it remains standard practice in LLM settings. While a real issue, I think in this context it is not on its own a reason to reject.

Personally, my main remaining concern is that the rebuttal comments repeatedly identified that the method could be applied to settings outside of mathematical reasoning and the models originally considered:

>  Importantly, CDE is a general exploration method and is not specialized to math; it should in principle apply to other domains as well. We believe our current experiments already demonstrate the effectiveness of CDE through consistent gains across multiple math datasets and ablations. To further strengthen the paper, we are running additional experiments in a different domain with a different model family.

While additional experiments were conducted with a new model, none were reported on another domain...making me suspect that the experiments mentioned didn't work out well. While this is a concern, it is indeed standard practice in current RLVR papers to only consider mathematical reasoning tasks, and given that several such datasets were considered, I also do not think this on its own is reason to reject.

I am thus left of the opinion that, while the submission is not perfect, it does clear the bar for presentation at this ICLR. I encourage the authors to add details of whatever the new non-mathematical experiment was, along with an analysis of perhaps why it did not work so well, to the final version, as well as addressing the various other lingering reviewer points.

**Reviewer Concerns:**

Discussed above.

**Reviewer Scores:**

Many reviewers here did in fact update their scores; I think other concerns were addressed, and a full discussion may have led to further increases.

---

### Decision · Program_Chairs · 2026-01-26

Accept (Poster)